# Q(D)O-ES: Population-based Quality (Diversity) Optimisation for Post Hoc Ensemble Selection in AutoML

**Lennart Purucker**[1] **Lennart Schneider**[2,3] **Marie Anastacio**[5]
**Joeran Beel**[1] **Bernd Bischl**[2,3] **Holger Hoos**[4,5,6]

[1]University of Siegen, [2]LMU Munich, [3]Munich Center for Machine Learning (MCML),
[4]Leiden University, [5]RWTH Aachen, [6]University of British Columbia

**Abstract** Automated machine learning (AutoML) systems commonly ensemble models post hoc to improve predictive performance, typically via greedy ensemble selection (GES). However, we believe that GES may not always be optimal, as it performs a simple deterministic greedy search. In this work, we introduce two novel population-based ensemble selection methods, QO-ES and QDO-ES, and compare them to GES. While QO-ES optimises solely for predictive performance, QDO-ES also considers the diversity of ensembles within the population, maintaining a diverse set of well-performing ensembles during optimisation based on ideas of quality diversity optimisation. The methods are evaluated using 71 classification datasets from the AutoML benchmark, demonstrating that QO-ES and QDO-ES often outrank GES, albeit only statistically significant on validation data. Our results further suggest that diversity can be beneficial for post hoc ensembling but also increases the risk of overfitting.

## 1 Introduction

Many automated machine learning (AutoML) systems do not return a single model but rather an ensemble of models. Following the taxonomy of ensembles from Cruz et al. (2018), AutoML systems perform three ensemble steps: *Generation*, the system generates *base models* while searching for an optimal configuration (*e.g.*, Auto-Sklearn (Feurer et al., 2015, 2022)) or cross-validating predetermined configurations (*e.g.*, AutoGluon (Erickson et al., 2020)); *Selection*, they select a subset of the base models generated in the first step, *e.g.* the top 50 models; *Aggregation*, they employ *post hoc ensembling* to aggregate the prediction of all selected base models.

Looking at ten prominent open-source AutoML systems, 60% rely on *post hoc* ensembling: Auto-Sklearn 1 (Feurer et al., 2015), Auto-Sklearn 2 (Feurer et al., 2022), AutoGluon (Erickson et al., 2020), Auto-PyTorch (Mendoza et al., 2018; Zimmer et al., 2021), H2O AutoML (LeDell and Poirier, 2020), and MLJAR (Płońska and Płoński, 2021). Among them, only H2O AutoML uses *stacking* (Wolpert, 1992) with a linear model, while all others rely on *greedy ensemble selection with replacement* (GES) (Caruana et al., 2004, 2006). The four systems that do not use *post hoc* ensembling by default – TPOT (Olson et al., 2016), GAMA (Gijsbers and Vanschoren, 2019), FLAML (Wang et al., 2021), and LightAutoML (Vakhrushev et al., 2021) – return the single best model; nevertheless, all of them but TPOT offer *post hoc* ensembling as an option.

The ten systems never studied the potential of *post hoc* ensembling nor evaluated different methods in their publications. The most frequently used method, GES, searches in a greedy manner for an optimal weight vector to linearly aggregate the predictions of base models. We believe, as GES uses a simple deterministic greedy search, there is potential to improve upon GES.

To do so, we focused on population-based optimisation methods as they have already shown success in optimisation (Kennedy and Eberhart, 1995; Das and Suganthan, 2010; Hansen and Auger, 2014; Kochenderfer and Wheeler, 2019). In particular, such methods allow us to add stochasticity while building the ensemble, *e.g.*, through crossover or random sampling. In contrast, GES performs

greedy improvement at each search step and thus cannot take advantage of interactions between base models that do not immediately improve performance; which might lead into local optima.

This motivated us to explore population-based ensemble selection with replacement for AutoML.

It is generally accepted in the literature that the diversity of an ensemble, from now on called *ensemble diversity*, can improve its performance (Dietterich, 2000b; Kuncheva and Whitaker, 2003; Sagi and Rokach, 2018). Prior work on ensemble selection *without* replacement, where one searches for an optimal subset of base models instead of a weight vector, incorporated ensemble diversity in the search (Partridge and Yates, 1996; Banfield et al., 2005; Partalas et al., 2010; Li et al., 2012). This motivated us to also include ensemble diversity in our optimisation method.

In the context of machine learning, ensemble diversity can be understood as a measure of the degree to which base models make different errors (Hansen and Salamon, 1990; Dietterich, 2000a; Banfield et al., 2005; Kumar and Kumar, 2012; Wood et al., 2023). For a given objective function or task, however, there often seems to be a trade-off between ensemble diversity and performance (Tang et al., 2006; Ahmed et al., 2017; Wood et al., 2023). This suggests that approaches such as that by Li et al. (2012), which jointly optimise ensemble diversity and performance, may be ineffective. For this reason, in our work, we focus on quality diversity optimisation (QDO).

QDO (Cully et al., 2015; Mouret and Clune, 2015; Chatzilygeroudis et al., 2021) is a recent trend in population-based optimisation. To avoid confusion, we note that the term *diversity* in QDO does not refer to ensemble diversity, but rather to *behavioural diversity*, i.e., the variability in behaviour between members of a given population. QDO maximises a single-objective function while maintaining a behaviourally diverse population. In robotics, for example, we could be interested in constructing bipedal robots with varying behaviour, e.g., tall and heavy or small and lightweight ones, that all should be able to walk fast in a straight line (Mouret and Clune, 2015).

With QDO for ensemble selection, we propose to maximise ensemble performance while maintaining a population of ensembles that is behaviourally diverse w.r.t. ensemble diversity. In other words, we maintain a population of top-performing ensembles with varying amounts of ensemble diversity. In contrast, a traditional population-based search keeps only the best-performing ensembles in the population – regardless of their ensemble diversity. As a behaviourally diverse population may be beneficial during optimisation (Mouret and Clune, 2015; Nguyen et al., 2015; Lehman and Stanley, 2011; Chatzilygeroudis et al., 2021), and ensemble diversity may be beneficial for performance (Sagi and Rokach, 2018), we see considerable promise in also considering QDO in our exploration of population-based ensemble selection.

Our contribution is the empirical performance comparison of GES against population-based search with and without diversity (QDO-ES and QO-ES, where QO stands for quality optimisation) for *post hoc* ensemble selection in AutoML. We show that population-based methods often rank better than GES on 71 classification datasets from the AutoML benchmark (Gijsbers et al., 2022). Moreover, we found that QDO-ES and QO-ES statistically significantly outperform GES on validation data, although the significance does not generalise to test data.

Our code and data are publicly available: see Appendix H.

## 2 Background

The goal of *post hoc ensembling* for AutoML is to aggregate a pool $P = \{p_1, ..., p_m\}$ of $m$ base models consisting of all models that are trained and validated during model selection or a subset thereof. For example, Auto-Sklearn sets $P$ to the 50 best models according to the validation score. The ensemble is trained on the models' predictions generated during validation, *i.e.*, the predictions on hold-out validation data or the out-of-fold predictions of $k$-fold cross-validation. Thereby, *post hoc* ensembling methods optimise a *user-defined* objective function. For the rest of this paper, we focus on classification, but the concept explained and our method can be extended to regression.

Formally, ensemble selection with replacement minimises an *ensemble loss function* $L(E)$ on the validation data, where $E$ is a multiset of base models, *i.e.*, a set that allows repetition, which can

be written as $E = (P, r)$, with $r \colon P \to \mathbb{Z}_0^+$ and $r(i)$ denotes the number of times base model $p_i$ is repeated in $E$. Given an ensemble $E$, we can compute its weight vector

$$w_E = \left[ \frac{r(i)}{\sum_{j=1}^{m} r(j)} \,\middle|\, i \in [1 \ldots m] \right]. \tag{1}$$

The prediction of $E$ is the $w_E$-weighted arithmetic mean of the prediction probabilities of all base models. The class with the highest prediction probability is predicted.

Greedy ensemble selection with replacement (GES) (Caruana et al., 2004, 2006) iteratively builds $E$ by means of a greedy deterministic search process that adds one base model at each iteration. GES is an ensemble selection method, also called *ensemble pruning* (Tsoumakas et al., 2009). Hence, it produces a sparse $w_E$ by design. Zero-weighted base models can be removed from the ensemble and increase neither inference time nor ensemble size; making both smaller than in non-sparse *ensemble weighting* methods, such as stacking. Algorithm 1 corresponds to the original definition of GES and has been implemented as such in AutoGluon. Auto-Sklearn omitted line 8, which means that the resulting ensemble might not be the one with the best performance *on the validation data*. The approach to break ties at lines 5 and 8 is also specific to the implementation.

---

**Algorithm 1** Greedy Ensemble Selection with Replacement

---

**Input**: Pool of base models $P$, ensemble loss function $L$, number of iterations $I$
**Output**: Weight vector $w$ of length $|P|$
1: $r \leftarrow [0 \cdots 0]$          ▷ Initialise the empty ensemble $E = (P, r)$.
2: $H \leftarrow \{r\}$          ▷ Initialise the ensemble history.
3: **for** $1 \ldots I$ **do**
4:      $R \leftarrow \{r' \mid r' = r$ with one element incremented by $1\}$     ▷ All possible repetitions of base model for this iteration.
5:      $r \leftarrow pickOne(\arg\min_{r' \in R} L((P, r')))$     ▷ *Select* the repetition(s) minimising the loss and pick one (to break ties).
6:      $H \leftarrow H \cup \{r\}$
7: **end for**
8: $r^* \leftarrow pickOne(\arg\min_{r' \in H} L((P, r')))$          ▷ Take the best seen ensemble as final ensemble.
9: **return** $w$ computed with Equation 1 using $E = (P, r^*)$.

---

## 3 Related Work

GES was first introduced to AutoML by Auto-Sklearn 1 (Feurer et al., 2015). The authors *stated* that GES outperforms alternatives such as stacking or gradient-free numerical optimisation. Recently, Purucker and Beel (2022) have shown that GES can perform better than other *post hoc* ensembling methods for data from OpenML (Vanschoren et al., 2013). Otherwise, to the best of our knowledge, GES was never compared to any other *post hoc* ensembling method.

Following Auto-Sklearn, most AutoML systems chose to employ GES as well; Auto-PyTorch and AutoGluon even based their initial implementation on the one from Auto-Sklearn. Aside from minor variations (*e.g.* in tie-breaking or computing the final weights), in all these cases, the underlying algorithm follows GES as defined by Caruana et al. (2004).

Population-based *alternatives* to GES have been explored in other fields, *e.g.*, to find a subset of base models for an ensemble (Partridge and Yates, 1996; Zhou et al., 2002; Zhou and Tang, 2003; Cavalcanti et al., 2016; Onan et al., 2017). QDO has been used previously to build ensembles (Boisvert and Sheppard, 2021; Nickerson and Hu, 2021; Cardoso et al., 2021b,a, 2022; Ferigo et al., 2023), but always with a focus on maintaining a behaviourally diverse *population of base models*, from which to build an ensemble after optimisation. In contrast, we focus on the behavioural diversity of a *population of ensembles*. Our QDO approach relates more closely to multi-objective optimisation for ensemble selection with quality and diversity as objectives (Partridge and Yates, 1996; Martınez-Munoz and Suárez, 2004; Banfield et al., 2005; Partalas et al., 2010; Li et al., 2012; Cavalcanti et al.,

2016). However, multi-objective approaches optimise both objectives, while QDO optimises only performance, benefiting from a behaviourally diverse population during optimisation.

## 4  Methods: Population-based Quality (Diversity) Optimisation for Ensemble Selection

Our methods maintain a population of ensembles to perform a stochastic search for an optimal weight vector. Following the concepts of ensemble selection *with replacement*, as introduced by GES, the final weight vector is sparse by design, and we express an ensemble as a multiset of base models $E$. We distinguish between quality optimisation for ensemble selection (QO-ES) and quality diversity optimisation for ensemble selection (QDO-ES) based on how the population is maintained.

Following QDO terminology, we store the population in an *archive A*, a set of size $a$. We build on pyribs (Tjanaka et al., 2021), a Python library for QDO, to implement archives. In QO-ES, we maintain the population by simply storing the $a$ observed solutions with the lowest loss in $A$. In contrast, to maintain a population for QDO-ES we additionally require a notion of behavioural diversity and a storage mechanism for $A$ that considers behavioural diversity.

In the following, we detail first how we maintain behavioural diversity in QDO-ES, and then the ways in which stochastic decisions are used in our approaches.

### 4.1  Maintaining Populations with Behavioural Diversity

QDO-ES requires a behaviour space $\mathcal{B}$ for *ensemble diversity*, such that we can determine the behavioural diversity w.r.t. ensemble diversity, and an archive that considers behavioural diversity.

#### 4.1.1  A Behaviour Space for Ensemble Diversity. We map an ensemble $E$ to a behaviour space $\mathcal{B}$ according to two ensemble diversity metrics:

*Average loss correlation (ALC)* measures the *explicit* ensemble diversity following previous work on correlation-based ensemble diversity metrics (Tumer and Ghosh, 1999; Brown et al., 2005). Our implementation measures the average Pearson correlation between loss vectors over all pairs of non-zero weighted base models in $E$. A loss vector contains the difference between 1 and the prediction probability of the correct class for each instance.

*Configuration space similarity (CSS)* measures the *implicit* ensemble diversity by measuring the average pairwise similarity of configurations of base models included in $E$ using the Gower similarity (Gower, 1971); it is implicit, since different configurations do not guarantee different predictions. To the best of our knowledge, the similarity of configurations has never been used in an ensembling context, but implicitly measuring the behavioural diversity has been used successfully in QDO for reinforcement learning (Ferigo et al., 2023). Moreover, CSS does not require a potentially biased definition of ensemble diversity, because it measures an existing variation in the *input space of algorithms* that produce the base models; in comparison, ALC measures the ensemble diversity in the *output space of the base models produced by these algorithms*.

ALC and CSS are formalised in Appendix D.1. We choose a two-dimensional behaviour space, because these are known to work well when behavioural diversity is not aligned with performance (Pugh et al., 2016). As pointed out in the introduction, there appears to be a trade-off between ensemble diversity and performance. Thus, the alignment of behavioural diversity and performance of an ensemble is generally unknown.

#### 4.1.2  An Archive for QDO-ES. In a typical QDO application, we divide the behaviour space *once* into $a$ niches (sometimes called partitions), represented by bins. The niches are computed based on the *theoretical range* of $\mathcal{B}$'s dimensions such that the niches' boundaries are uniformly distributed across $\mathcal{B}$. During optimisation, the best-observed solutions for each niche are then stored in the archive $A$. In detail, solutions are stored in the niche corresponding to their behaviour values if their loss is smaller than the current solution in the bin (or if the bin is empty). Thus, $A$ in QDO enforces local competition between behaviourally similar solutions (Chatzilygeroudis et al., 2021). We used

a *sliding boundaries archive* (Fontaine et al., 2019) for QDO-ES to enable better local competition and more representative random sampling; motivated in more detail in Appendix D.2.

Note that QDO practitioners are interested in the best solutions for all niches, while we, in AutoML, are only interested in the single best ensemble. Our application of QDO is atypical; nevertheless, having access to a diverse population during optimisation can improve performance, even if one is only interested in a single best solution (Nguyen et al., 2015; Mouret and Clune, 2015).

## 4.2 Stochasticity during Optimisation

We implemented three ways to include stochasticity during optimisation: sampling of parents, crossover, and mutation. We sample two solutions to apply crossover (potentially followed by mutation) on them; alternatively, if no crossover is applied, we sample only one and mutate it.

### 4.2.1 Sampling. We implemented three approaches to sample parents from an archive: *deterministic sampling*, which returns the best solution(s) from the archive; a non-deterministic variant of *tournament selection* (Miller and Goldberg, 1995), which samples a set of solutions and returns the winner(s) of a randomized tournament – described detailed in Appendix D.3; and *dynamic sampling*, an adaptive approach that either uses deterministic or random sampling.

In dynamic sampling, the initial probability of random sampling is 50% such that both choices are equally likely in the first iteration. The probability of deterministic *vs.* random sampling is updated after each iteration, based on the ratio between the average performance of both sampling approaches over a window of recent iterations, such that the sampling strategy with higher average performance is more likely to be used in the next iteration; see Appendix D.4.

We opted for an adaptive sampling approach because we expect the optimal probability of deterministic *vs.* random sampling changes over time – like an exploration-exploitation tradeoff (Qin and Suganthan, 2005; Audibert et al., 2009; Li et al., 2013). Furthermore, we did not want to introduce an additional hyperparameter.

### 4.2.2 Mutation. To keep the resulting weight vector sparse and to adhere to the concepts of ensemble selection with replacement, we follow the idea of GES, in the sense that we adjust $r$ during mutation by incrementing one of its elements. We introduce additional stochasticity by randomly choosing this element. We provide more details and pseudo-code on the mutation operator in Appendix D.5.

### 4.2.3 Crossover. We use two-point crossover (Jong and Spears, 1992) or average crossover (Li et al., 2013). We apply crossover to $r'$ and $r''$ for two ensembles $E'$ and $E''$ instead of $w_{E'}$ and $w_{E''}$; otherwise, crossover could produce a vector for which we cannot produce an $r$ for the multiset $E$, *i.e.*, compute the reverse of Equation 1. To produce a valid multiset, we round the results of average crossover to the next higher integer. For two-point crossover, we observed that the offspring were often almost only zeros, because $r'$ and $r''$ are sparse. To counteract this, we apply two-point crossover only on the elements that are non-zero in $r'$ or $r''$, see Appendix D.6 for more details.

The probability of using crossover adapts through the run, similar to the adaptive sampling approach. Following the same mechanism, the offspring produced by crossover may mutate.

## 4.3 Putting Everything Together

The final realisation of Q(D)O-ES is described in Algorithm 2. First, $initArchive(A, P)$ tries to insert an initial set of ensembles into the archive (Algorithm 2, line 1). We implemented three approaches initialising the set: the ensembles consisting only of a base model, all ensembles of size 2 including the best base model, or $m$-many random ensembles of size 2, formalised in Appendix D.7.

Second, a batch of solutions is built in each iteration (Lines 4-11) using $sample()$, $crossover()$, and $mutate()$. The inner workings of each of those functions depend on their hyperparameters and current adaptive probability. In the end, a new solution is created by only crossover, only mutation, or crossover and mutation. Finally, we evaluate the proposed solutions contained in the batch and

try to insert them into the archive (Lines 12). Due to randomness, proposed solutions may equal previously evaluated ones. Hence, we introduced rejection sampling (Algorithm 2, line 8) such that previously evaluated ensembles are not added to the batch. To avoid endless rejection sampling in observed edge cases, we added an emergency brake to stop the loop, see Appendix D.8.

Finally, we compute $r^*$ using $bestSolution(A)$, which returns the best $r$ according to the validation loss among: the single best model, the average crossover of all solutions in $A$, and the best solution in $A$.

---

**Algorithm 2** Population-based Quality (Diversity) Optimisation for Ensemble Selection

---

**Input**: Pool of base models $P$, ensemble loss function $L$, quality (diversity) archive $A$, number of iterations $I$, batch size $B$
**Output**: Weight vector $w$ of length $|P|$
 1: $A \leftarrow initArchive(A, P)$          ▷ Fill the archive with a set of initial ensembles.
 2: **for** $1 \ldots I$ **do**
 3:     $S \leftarrow \emptyset$
 4:     **while** $|S| < B$ **do**          ▷ Build the batch $S$.
 5:        $r, r' \leftarrow sample(A)$          ▷ If crossover is deactivated, $r = r'$.
 6:        $r_{sol} \leftarrow crossover(r, r')$
 7:        $r_{sol} \leftarrow mutate((P, r_{sol}))$          ▷ If mutate is deactivated, $r_{sol} = mutate((P, r_{sol}))$.
 8:        **if** $unknown(r_{sol})$ **then**          ▷ Rejection sampling.
 9:           $S \leftarrow S \cup r_{sol}$
10:        **end if**
11:     **end while**
12:     $A \leftarrow insertIntoArchive(A, S)$          ▷ Add solutions to the archive and update the boundaries for QDO-ES.
13: **end for**
14: $r^* \leftarrow getBestSolution(A)$
15: **return** $w$ computed with Equation 1 using $E = (P, r^*)$.

---

## 5 Experiments

We compare GES, QO-ES, and QDO-ES. Moreover, we used the single best model as a baseline.

The datasets used in our experiments and the final evaluation follow the evaluation procedure described and discussed in the AutoML benchmark (AMLB) (Gijsbers et al., 2022). We follow the AMLB because, in our experiments, the output of an ensemble method simulates the output of an AutoML system as if it would have used the evaluated method for *post hoc* ensembling.

More precisely, we evaluated the methods w.r.t. ROC AUC using 10-fold cross-validation on the 71 OpenML classification datasets used in the AMLB. For multi-class, we use macro average one-vs-rest ROC AUC. Since ROC AUC requires prediction probabilities and is independent of a decision threshold, we complement it by also evaluating w.r.t. balanced accuracy, which requires predicted labels and is dependent on a threshold (rather than another threshold-independent metric such as log loss). In accordance to the AMLB procedure, we distinguish between binary and multi-class. We refer to a combination of a classification setting with an evaluation metric as a *scenario* (e.g., ROC AUC for multi-class).

**Base Model Generation**. To simulate *post hoc* ensembling methods for AutoML, we require a pool of base models $P$ as generated by an AutoML system. Therefore, we ran Auto-Sklearn 1 on each fold of each dataset, twice – once for each metric. For each fold, we stored the set of all models validated during model selection. In our experiment and by default, Auto-Sklearn 1 uses a 33% hold-out split from the training data as validation data. Afterwards, we pruned this set per fold to 50. As pruning is a preprocessing step before *post hoc* ensembling, we decided to include two pruning strategies in our experiments, denoted as *TopN* and *SiloTopN*, such that we obtain two *Ps* per fold. For *TopN*, we pruned to the top 50 best-performing models according to the validation score following Auto-Sklearn's default behaviour. For *SiloTopN*, we pruned to 50, such that as many top-performing models of each algorithm family are kept following AutoGluon's approach.

Following the AMLB, we gave Auto-Sklearn a budget of 4 hours, 32 GB of memory, and 8 cores per fold. In all our experiments, we used AMD EPYC 7452 CPUs. We increased the memory to

64 or 128 GB, to prevent Auto-Sklearn from running out of memory for 11 datasets. As a result, Auto-Sklearn generated at least 50 models for all but 5 datasets, see Appendix E.1 for an overview.

**Hyperparameter settings**. Because the behaviour of QO-ES and QDO-ES depends heavily on their hyperparameters, and to make sure that all methods are compared fairly, we defined a grid of hyperparameter settings and exhaustively evaluated it. This led to the evaluation of 219 distinct configurations: 1 for the single best model; 2 for GES, one for AutoGluon's and one Auto-Sklearn's variant (see Section 2); 108 for QD-ES; and 108 for QDO-ES. We also treat the pruning strategy as a hyperparameter, which doubles our configuration space to 437 (the single best is identical for both pruning methods). The list of hyperparameter settings for Q(D)O-ES is available in Appendix E.2.

To compute the final evaluation score, the average over the folds, we ran each configuration for every fold of each dataset for both metrics – that is, in total, we evaluated $437 * 71 * 10 * 2 = 620540$ ensemble runs. We set the number of iterations to 50 for GES following Auto-Sklearn's default and guaranteed that Q(D)O-ES used the same number of total function evaluations, i.e., $50 * m$, by adjusting the number of iterations in Algorithm 2 depending on the batch size and having one remainder batch if needed. To measure the efficiency, we capture the running time of the ensemble methods to complete the 50 iterations or $50 * m$ function evaluations. Moreover, we capture the size of the final ensemble by counting how many base models are non-zero weighted and therefore influence the final prediction. We implemented multiprocessing for GES and Q(D)O-ES and ran all configurations with the same hardware and resources used for Auto-Sklearn.

To select a configuration for each method on each dataset, we used leave-one-out cross-validation (LOO CV). For $d$ datasets, the configuration with the highest *median normalised improvement* on $d - 1$ datasets is selected to represent the method on the out-of-fold dataset. A discussion on why we used the median normalised improvement and find it to be the only appropriate method for our experiments can be found in Appendix E.3.

**Normalised Improvement**. We use normalised improvement, following the AMLB. Thereby, we scale the scores per dataset, such that 0 is equal to the performance of the best configuration on this dataset and $-1$ is equal to the performance of the single best model. If their performance is equal, then we set everything as good as the single best model to $-1$ and penalize worse configurations to $-10$, see Appendix E.4. For the LOO CV selection, we only normalise across configurations from the same method, while we normalise across all selected configurations for the final evaluation.

**Statistical Tests**. Again following the AMLB, we perform a Friedman test with a Nemenyi *post hoc* test ($\alpha = 0.05$) on the non-normalised scores of each method to obtain an absolute ranking of all methods and to test for statistically significant differences between them.

## 6 Results

For each scenario, we visualise the mean absolute rank and results of the statistical tests via critical differences in Figure 1. We observe that for all scenarios, *post hoc* ensemble selection is statistically significantly better than the single best model – we can always boost the performance on average. Q(D)O-ES ranks higher than GES in all scenarios, except for balanced accuracy multi-class.

To further investigate our results, we use boxplots of the normalised improvement to visualise the distribution of the relative performance for all methods over the datasets – see Figure 2. Even though the ensemble methods outperform the single best (indicated by a red line) on average, there are datasets on which this does not hold (as indicated by the numbers in the square brackets in Figure 2). Because all methods have access to the validation score of the single best and only propose an ensemble if it outperforms the single best, this phenomenon is directly linked to overfitting.

We want to note that computing all results without parallelization across datasets, folds, configurations, and metrics but with parallelization of the AutoML systems and the ensemble method on 8 cores took approximately around 3.9 years of wall-clock time, see Appendix F.1.

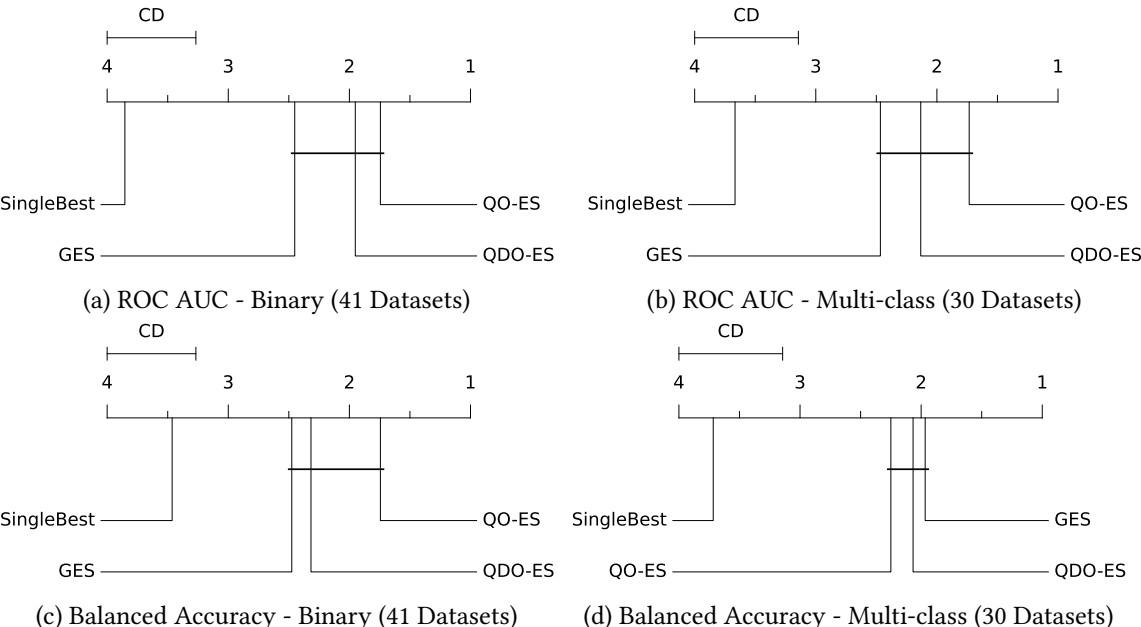

(a) ROC AUC - Binary (41 Datasets)

(b) ROC AUC - Multi-class (30 Datasets)

(c) Balanced Accuracy - Binary (41 Datasets)

(d) Balanced Accuracy - Multi-class (30 Datasets)

Figure 1: **Critical Difference Plots for Test Scores**: Mean rank of the methods (lower is better). Methods connected by a bar are not significantly different.

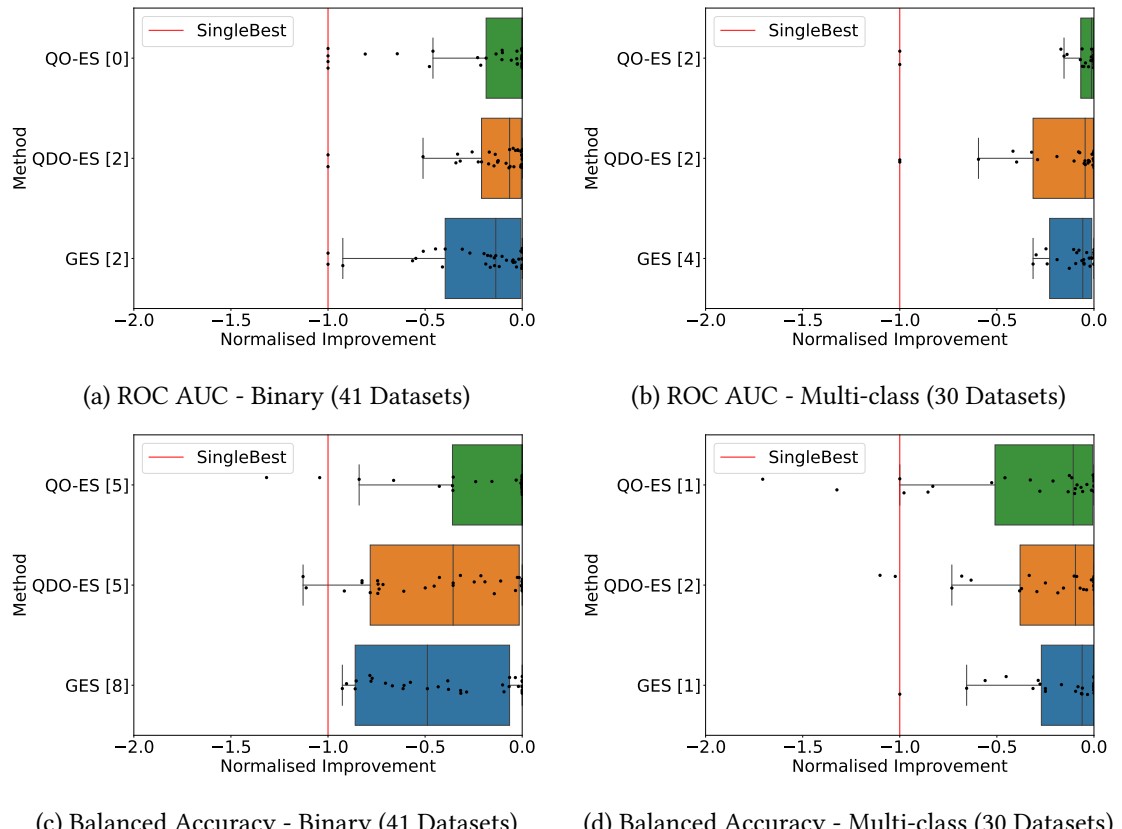

(a) ROC AUC - Binary (41 Datasets)

(b) ROC AUC - Multi-class (30 Datasets)

(c) Balanced Accuracy - Binary (41 Datasets)

(d) Balanced Accuracy - Multi-class (30 Datasets)

Figure 2: **Normalised Improvement Boxplots**: Higher is better. Each dot represents a dataset. The number in square brackets next to a method's name counts the outliers smaller than −2.

**Overfitting**. We observed that overfitting had a large influence on the final testing performance of our methods when repeating our evaluation for validation instead of testing scores; see Figure 3. We note that the difference between Q(D)O-ES and GES is statistically significant in all scenarios. Moreover, the normalised improvement gains for Q(D)O-ES are also substantially larger than for GES, see Appendix F.2. Likewise, as expected, no *post hoc* ensembling method performs worse than the single best model on any dataset. Thus, the main challenge *post hoc* ensembling encounters is the lack of generalisation from validation to testing sets.

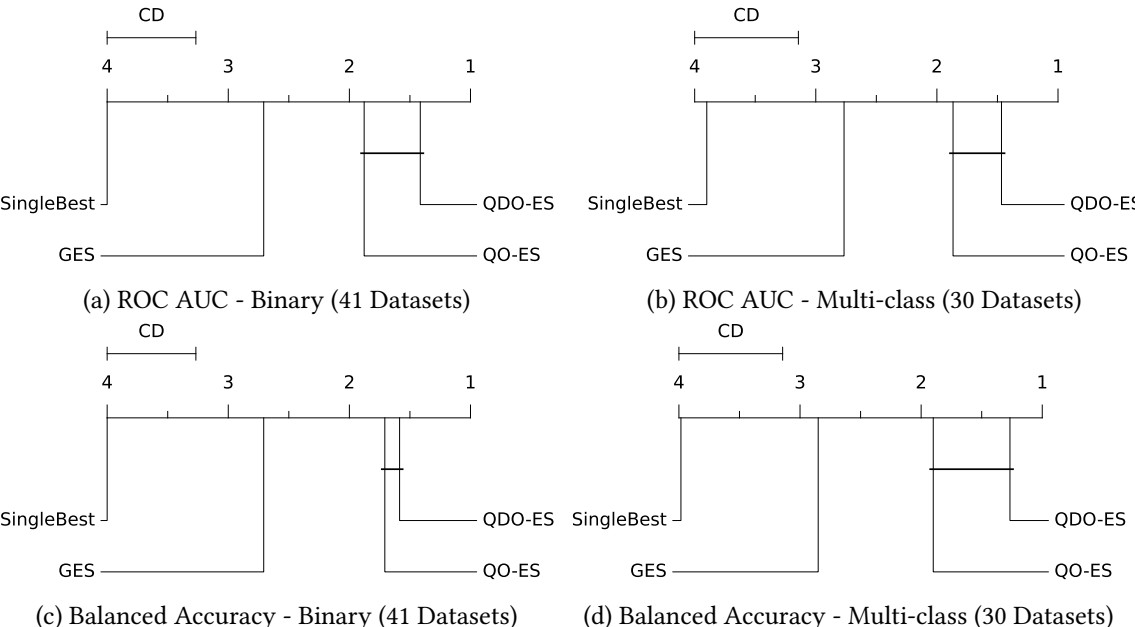

(a) ROC AUC - Binary (41 Datasets)  (b) ROC AUC - Multi-class (30 Datasets)

(c) Balanced Accuracy - Binary (41 Datasets)  (d) Balanced Accuracy - Multi-class (30 Datasets)

Figure 3: **Critical Difference Plots for Validation Scores**: Mean rank of the methods (lower is better). Methods connected by a bar are not significantly different.

We believe that the validation data generated by Auto-Sklearn's 33% hold-out split may be responsible for the large difference between validation and test performance. In future work, it would be interesting to investigate the performance and generalisation of our methods for other AutoML systems that use more sophisticated validation procedures, *e.g.*, AutoGluon with n-repeated k-fold cross-validation.

Regarding GES, we observe that based on the LOO CV procedure, AutoGluon's variant is always selected for balanced accuracy, while for ROC AUC, Auto-Sklearn's variant is selected for all multi-class classification datasets and ~12% (5/41) binary classification datasets. For validation scores, AutoGluon's variant is always selected. We note that the difference in the implementations of AutoGluon and Auto-Sklearn (see Section 2) can be interpreted as differences in handling overfitting: the first variant is more prone to overfitting than the latter, due to relying more on the validation score for the selection of the final ensemble.

**Ensemble Diversity**. We note that for all methods, the preprocessing approach selected based on the LOO CV procedure was SiloTopN, rather than TopN. The diversity of base models fostered by including algorithms from each family seems to always be beneficial. This difference in algorithmic diversity is also visible in the pool of base models $P$ per dataset. The average number of distinct algorithms in $P$ across all datasets and metrics is ~2.77 for TopN while SiloTopN achieves ~15.33; see Table 2 in Appendix E.1 for numbers per dataset. We also observe this when analysing hyperparameter importance, see Appendix F.3; we found the preprocessing method to be the most important hyperparameter, followed by the sampling and initialisation method.

Table 1: **Ensemble Efficiency**: Average ensemble size and running time for 50 iterations in seconds.

| | Average Ensemble Size | | | | Average Running Time @50 | | | |
| | Balanced Accuracy | | ROC AUC | | Balanced Accuracy | | ROC AUC | |
| Method | Binary | Multi-class | Binary | Multi-class | Binary | Multi-class | Binary | Multi-class |
|---|---|---|---|---|---|---|---|---|
| GES | 7.95 | 7.91 | 8.02 | 10.09 | 49.13 | 120.36 | 43.66 | 197.57 |
| QDO-ES | 9.23 | 11.47 | 14.06 | 13.27 | 81.92 | 158.24 | 91.04 | 451.48 |
| QO-ES | 8.02 | 10.28 | 15.76 | 13.88 | 65.41 | 132.76 | 75.5 | 388.62 |

Regarding the use of ensemble diversity for optimisation, we see that QO-ES beats QDO-ES except for balanced accuracy multi-class on testing data. On validation data, QDO-ES always beats QO-ES, although the differences in rank are not significant. We hypothesise that adding ensemble diversity to the optimisation process can, in principle, be beneficial, but also increases the risk of overfitting, because it utilises the distribution of ensemble diversity, which might not generalise.

**Efficiency**. See Table 1 for an overview of the efficiency for each ensemble method per scenario. The size of an ensemble corresponds to how many base models must compute predictions and thus directly represents the inference efficiency of an ensemble. To investigate the optimisation efficiency of the methods we studied, we looked at the average running time required for completing 50 iterations. Note that for multi-class, the methods are more efficient for balanced accuracy than for ROC AUC, as computing balanced accuracy is more costly than ROC AUC for multi-class.

The average size across all scenarios of the generated ensembles is ~8.5 for GES, ~12 for QO-ES, and ~12 for QDO-ES. Although Q(D)O-ES generates slightly bigger ensembles, their weight vectors are still sparse, considering that 50 base models were available in most cases. GES needs on average across all scenarios ~103 seconds. Our current implementation of QO-ES takes ~165 seconds, and QDO-ES ~196 seconds, but optimising our code would likely increase efficiency. We note that these times are insignificant compared to the time budget of 4 *hours* expended by Auto-Sklearn for finding the base models.

**Ablation Study**. When performing ablation studies for Q(D)O-ES, we found that the preprocessing approach seems to be the most important component for our approach; followed by the sampling method and approach of archive initialisation (see Appendix F.3). Specifically, SiloTopN seems to always perform better than TopN, as indicated by the selected preprocessing approach above (see Appendix F.4). The best sampling and initialisation method seem to differ slightly per scenario.

## 7 Conclusion

*Post hoc* ensembling is widely used in AutoML systems and can be crucial for obtaining peak predictive performance. One of the most popular methods is greedy ensemble selection with replacement (GES) (Caruana et al., 2004, 2006). In this work, we presented *QO-ES* and *QDO-ES*, two novel population-based optimisation algorithms for *post hoc* ensemble selection in AutoML. QO-ES maintains an archive of the best-performing ensembles and sequentially improves upon them, making use of mutation and crossover operators. QDO-ES builds upon QO-ES, but leverages concepts from quality diversity optimisation (Chatzilygeroudis et al., 2021), maintaining an archive of differently behaviourally diverse ensembles during optimisation. In extensive experiments, we demonstrated that 1) *post hoc* ensemble selection improves upon the single best model, 2) our population-based methods QO-ES and QDO-ES generally outperform GES (although the performance differences are not statistically significant on testing data), and 3) respecting the diversity of ensembles during optimisation can be beneficial but also increases the risk of overfitting. Lastly, we want to highlight that overfitting is a serious challenge for *post hoc* ensembling in AutoML.

**Acknowledgements**. The CPU nodes of the OMNI cluster of the University of Siegen (North Rhine-Westphalia, Germany) were used for all experiments presented in this work. This research is partially supported by the Bavarian Ministry of Economic Affairs, Regional Development and Energy through the Center for Analytics - Data - Applications (ADACenter) within the framework of BAYERN DIGITAL II (20-3410-2-9-8) and by TAILOR, which is part of the EU Horizon 2020 research and innovation programme under GA No 952215. Finally, we thank the reviewers for their constructive feedback and contribution to improving the paper.

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

## B  Limitations

We note that our work is limited with respect to the following points: 1) we found that overfitting is a large problem but only were able to compare our methods for Auto-Sklearn 1 and not additional AutoML systems that have better validation data available, such as Auto-Sklearn 2 or AutoGluon; 2) we were only able to explore a subset of all possible hyperparameters and hyperparameter settings for our newly proposed methods; 3) we were only able to evaluate w.r.t. two metrics; and 4) we were not able to explore all potential variations of QDO that could have been used for QDO-ES.

## C  Broader Impact Statement

We believe that our work is mostly abstract and methodical. Thus, after careful reflection, we determined that this work presents no notable or new negative impacts to society or the environment that are not already present for existing state-of-the-art AutoML systems. We proposed to replace one component of an AutoML system such that the predictive performance improves while efficiency stays mostly the same. Therefore, we only see the potential positive impact that higher predictive performance might help to make better decisions using AutoML.

## D  Supplements for the Algorithm Description

### D.1  Formula for Diversity Measures

To measure the diversity of an ensemble $E = (P, r)$, we consider all non-zero weighted base models in the ensemble, that is, $P' = \{p_i \in P \mid r(i) > 0\}$ with $|P'| = m'$.

The average loss correlation (ALC) is defined for $P'$ and a set of loss vectors $L = \{l_1, ..., l_{m'}\}$. A loss vector $l_i$ corresponds to the difference between 1 and the prediction probability of $p_i$ for the correct class for each instance. Then, ALC is defined as the average Pearson correlation $\rho$ of two loss vectors over all pairs of models in $P'$:

$$ALC = \frac{2}{m'(m'-1)} \sum_{i=1}^{m'-1} \sum_{j=i+1}^{m'} \rho(l_i, l_j). \tag{2}$$

The configuration space similarity (CSS) is defined for $P'$ and a set of configurations $C = \{c_1, ..., c_{m'}\}$ where $c_i$ is the configuration of $p_i$. We assume that the configurations are from the same configuration space such that at least the hyperparameter describing the used algorithm exists for any two $c$. Moreover, we assume that we are given the maximal observed ranges in $C$

of all numerical hyperparameters. For a numerical hyperparameter $h$, we denote its range with $R_h$. Generally, we denote the value of a configuration $c$ for a hyperparameter $h$ with $h^c$. CSS is the average Gower distance (Gower, 1971) over all pairs of models in $P'$:

$$CSS = \frac{2}{m'(m'-1)} \sum_{i=1}^{m'-1} \sum_{j=i+1}^{m'} (1 - G_{i,j}). \tag{3}$$

The Gower similarity $G_{i,j}$ is defined only for the set of hyperparameters $K = \{h_1, ..., h_{|K|}\}$ that appear in $c_i$ and $c_j$. Furthermore, the Gower similarity differentiates between categorical and numeric hyperparameters and is the average over the similarity of each hyperparameter in $K$:

$$G_{i,j} = \frac{1}{|K|} \sum_{k=1}^{|K|} \begin{cases} G_{i,j,k}^{\text{cat}}, & \text{if } h_k \text{ is categorical} \\ G_{i,j,k}^{\text{num}}, & \text{otherwise} \end{cases}, \tag{4}$$

with

$$G_{i,j,k}^{\text{cat}} = \begin{cases} 1, & \text{if } h_k^{c_i} = h_k^{c_j} \\ 0, & \text{otherwise} \end{cases}, \tag{5}$$

and,

$$G_{i,j,k}^{\text{num}} = \frac{|h_k^{c_i} - h_k^{c_j}|}{R_{h_k}}. \tag{6}$$

As at least the hyperparameter describing the used algorithm exits for both configurations, $G_{i,k} \geq 0$. Moreover, it is at most 1 if all hyperparameters are identical.

## D.2 Motivation to use a Sliding Boundaries Archive for QDO-ES

For QDO-ES, we implemented a *sliding boundaries archive* (Fontaine et al., 2019). A sliding boundaries archive regularly recomputes the niche boundaries based on the *observed range* of behaviour values after initialising the niches based on the theoretical range.

We use a sliding boundaries archive because we cannot guarantee that the *observable* values of a behaviour space for ensemble diversity are uniformly distributed, as is usually assumed for QDO (Chatzilygeroudis et al., 2021; Nickerson and Hu, 2021). Moreover, the observable range of ensemble diversity metrics differs between datasets since they depend, *e.g.*, on the base models' predictions. The sliding boundaries archive allows us to maintain a similar granularity of the partitions between datasets. The granularity will also be finer than the initially computed partitions and thus enable more local competition.

Additionally and more importantly, because we keep one ensemble per niche, the sliding boundaries archive aligns the population with the underlying distribution of the observed behaviour space, making random sampling more representative.

## D.3 Used Variant of Tournament Selection

In our non-deterministic variant of tournament selection, we define the tournament size $T$ based on the number of solutions we want to sample. If we want to sample one solution, we randomly sample 10 solutions from the archive. We sample 20 solutions if we want to sample two solutions. In the first few iterations, we might have less than $T$ solutions in the archive. In such a case, we take all solutions in the archive and additionally mutate solutions, taken from the initial set of solutions proposed to the archive, with Algorithm 3.

Given $T$-many solutions, we assign each solution a probability of being sampled. The best-performing solution is assigned a probability of 0.8, the second best $0.8 * 0.2^1$, the third best $0.8 * 0.2^2$, and so on until $0.8 * 0.2^{(T-1)}$. Finally, we sample either one or two solutions with these probabilities. If we sample two solutions, we sample without replacement.

### D.4 Implementation of an Adaptive Probability

We have several components in our algorithm where a probability defines whether these are activated or deactivated or how they are configured. In all cases, we used a self-adaptive probability instead of a predetermined constant value or function over time. Thus, avoiding additional hyperparameters and the bias of selecting a constant value or function over time for all tasks. This includes: dynamic sampling; whether crossover is used; and whether mutation is used after crossover.

We can represent such a choice as a binary decision between using $a$ or $b$. Initially, the probability of using $a$, $Pr(a)$, is set to 50% (consequently $Pr(b) = 1 - Pr(a)$). That is, we randomly select whether to use $a$ or $b$. Then, after each iteration, the probability is adapted based on the observed performance of $a$ and $b$ over the last $f$ iterations. $f$ is a hyperparameter. We set $f = 10$ because we aimed for a window over the performance such that bad performance in earlier iterations can be forgotten in face of dynamic changes to the performance of $a$ and $b$. We track the performance of solutions in relation to their origin, i.e., whether the solutions were produced with $a$ or $b$. Moreover, we do not change the probability if less than 2 solutions were created with $a$ or $b$.

For the update, we first compute the average performance over all solutions seen in the last 10 iterations for $a$ and $b$, denoted as $a_{avg}$ and $b_{avg}$. Using these values, we compute: $Pr(a) = 1 - \frac{a_{avg}}{a_{avg} + b_{avg}}$. Finally, we bound $Pr(a)$ in $[0.05, 0.95]$ for dynamic sampling and in $[0.1, 0.9]$ otherwise. We apply these bounds to avoid reaching a probability of 0 for $a$ or $b$ such that it would not be chosen anymore and consequently not have solutions for an update.

We have tighter bounds for mutation after crossover and crossover, since we observed that a probability of 0.05 behaved similarly to a probability of 0 – not enough solutions were produced such that future updates could change the probability. This happens only for mutation after crossover and crossover, as both are only used for a subset of all solutions. In contrast, dynamic sampling is always used as the initial step for creating a new solution (see Algorithm 2); assuming dynamic sampling is used and not another sampling method.

### D.5 Mutation Operator

Our mutation operator follows the idea of GES (Algorithm 1), in the sense that we adjust $r$ during mutation by incrementing one of its elements. We introduce additional stochasticity by randomly choosing this element. Consequently, we implemented a randomized version of GES's update as a mutation operator, see Algorithm 3. However, since $E$ is not built iteratively in Q(D)O-ES, we need to track which base model we previously added to avoid producing the same ensemble twice (handled by lines 2, 5, and 6).

---

**Algorithm 3** Iteration Independent Randomized With-replacement-like Mutation

**Input**: Ensemble $E = (P, r)$
**Output**: Ensemble $E_{mutated} = (P, r_{mutated})$
1: $R \leftarrow \{r' \mid r' = r$ with one element incremented by 1$\}$        ▷ Obtain all possible extensions of $r$
2: $R_{seen} \leftarrow lookUpSeenFor(r)$        ▷ Obtain previously produced extensions
3: $R_{potential} \leftarrow R \setminus R_{seen}$
4: $r_{mutated} \leftarrow randomSample(R_{potential})$
5: $R_{seen} \leftarrow R_{seen} \cup r_{mutated}$        ▷ Update previously produced extensions
6: $storeForLookUp(r, R_{seen}); storeForLookUp(r_{mutated}, \emptyset)$        ▷ Store previously produced extensions
7: **return** $(P, r_{mutated})$

---

### D.6 Two-Point Crossover of Repetition Vectors

For two-point crossover of two repetition vectors $r$ and $r'$, we first select the subsets $r_{sub}$ and $r'_{sub}$ of elements that are non-zero in either $r$ or $r'$. If this subset is smaller than three, we fall back to average crossover. We require a vector of at least length three to perform two-point crossover. Otherwise, only one possibility to perform two-point crossover would exist.

Next, we select two distinct points randomly and crossover $r_{sub}$ and $r'_{sub}$ with these points. Afterwards, we fill $r$ / $r'$ with the elements of $r_{sub}$ / $r'_{sub}$ creating $r_{co}$ / $r'_{co}$. Finally, we verify that the elements of $r_{co}$ / $r'_{co}$ are not all zero and return $r_{co}$ / $r'_{co}$. If neither $r_{co}$ nor $r'_{co}$ contain non-zero elements, then we again fall back to average crossover to produce offspring.

Two-point crossover can produce two unique children. If this happens, we add both (mutated) children to the batch in QO-ES and QDO-ES (Algorithm 2).

## D.7 Initialisation Approaches

Population-based search requires an initial population $S$. To produce the initial population for Q(D)O-ES, we implemented three approaches: all ensembles consisting only of a base model, all ensembles of size 2 including the best base model, or $m$-many random ensembles of size 2. Formally, the set of all ensembles consisting only of one base model is given by

$$S_1 = \left\{ (P, r) \;\middle|\; \sum_{i=1}^{m} r(i) = 1 \right\}. \tag{7}$$

The set of all ensembles of size 2 including the best base model with $j$ the index of the single best model, is given by

$$S_2 = \left\{ (P, r) \;\middle|\; r(j) = 1 \wedge \sum_{i=1}^{m} r(i) = 2 \right\}. \tag{8}$$

The single best model is the base model in $P$ with the highest validation score. To construct the set $S_3$ of $m$-many random ensembles of size 2, we randomly select two members of the ensemble $m$ times. Formally, $S_3 \subset \hat{S}_3$ with $|S_3| = m$ and $\hat{S}_3$ the set of all possible ensembles of size 2:

$$\hat{S}_3 = \left\{ (P, r) \;\middle|\; \exists! i, j \in \{1, \ldots, m\}, i \neq j : r(i) = 1 = r(j) \right\}. \tag{9}$$

Note, the size of this initial population differs between the methods by definition; $S_1 = S_3 = m$ while $S_2 = m - 1$.

## D.8 Emergency Brake for Rejection Sampling

We added an emergency brake that temporarily changes the increment during mutations (Algorithm 3, Line 1) and/or probability of crossover when more than 50 solutions are rejected in one iteration.

We were motivated to do this because we observed an edge case where Algorithm 2 was not able to propose any unseen solutions. Thus, the algorithm ran into an endless loop of rejection sampling. We observed this in the first few iterations of the algorithm after the archive was initialised with ensembles consisting only of a base model, or when the probability of crossover was very high in the first few iterations.

Our overall emergency brake consists of two parts, one brake for rejections resulting from mutation and one for rejections resulting from crossover. In one iteration, both brakes can be activated (multiple) times at the same time if the algorithm still does not produce unseen solutions.

If mutation does not produce any new solutions and crossover is deactivated (or does not produce new solutions either), then the mutation emergency brake is activated after 50 rejections resulting from mutation. Once the break is activated, the increment of the repetition vectors during mutation (Algorithm 3, Line 1) is increased by 1 for the current iteration.

The rejections might also result from solutions created by crossover and not by mutation because crossover cannot produce unseen children, and the adaptive probability of mutation after crossover is too small to be called in the current iteration. In this case, we increase the probability of mutation after crossover to 100% for this iteration after 50 rejections stemming from crossover.

## E  Supplements for the Experiments

### E.1  Dataset Overivew

Table 2 gives an overview of datasets and base models used in our experiments. Additionally, we show the average (over folds) for the number of base models and the number of distinct algorithms over the pools of base models for balanced accuracy $\mathcal{BA}$ and ROC AUC $\mathcal{R}$, respectively for the preprocessing methods *SiloTopN* and *TopN*.

### E.2  Configuration Space

Table 3 shows all hyperparameter and their possible values for the configuration space of QO-ES and QDO-ES. We refer to the set of all ensembles consisting only of a single base model as *L1 Ensembles*, while we refer to all ensembles of size 2 that include the single best model, or *m*-many random ensembles of size 2 as *L2 Ensembles*.

### E.3  Discussion on the Configuration Selection Approach

We use the median normalised improvement to select the most representative configuration during leave-one-out cross-validation because it reflects what we believe to be the distinguishing factor between different methods based on an evaluation following the AutoML benchmark (AMLB) (Gijsbers et al., 2022) – the highest median in the boxplots of the normalised improvement.

We are aware that alternative selection approaches could change our results or that one could, in principle, learn to select the best configuration. However, the only valid alternative in an evaluation following the AMLB would have been mean rank. As this is the only other distinguishing factor used in plots, specifically critical difference plots. However, as noted already in the AMLB and visible in its results, the absolute rank is not representative of a performance distribution and hides relative performance differences. In contrast, the median normalised improvement is based on a relative measure and relates to a performance distribution via the median.

### E.4  Normalised Improvement

Our implementation of normalised improvement follows the AutoML benchmark (Gijsbers et al., 2022). That is, we scale the scores for a dataset such that $-1$ is equal to the score of the single best model, and 0 is equal to the score of the best method on the dataset.

Formally, we normalise the score $s_D$ of a method for a dataset $D$ using:

$$\frac{s_D - s_D^b}{s_D^* - s_D^b} - 1, \tag{10}$$

with the score of the baseline $s_D^b$ and the best-observed score for the dataset $s_D^*$. We assume that higher scores are always better.

We extend this definition for the edge cases where no method is better than the baseline, *i.e.*, $s_D^* - s_D^b = 0$. We assume that this edge case never happened in the AutoML benchmark. Otherwise, their definition and implementation would have been undefined / crashed. In our setting, such an edge case can happen due to overfitting such that the ensemble methods becomes worse than the single best model.

We generalize the definition of this edge case to the case $|s_D^* - s_D^b| <= \Delta$. We set $\Delta$ to 0.0001, because normalised improvement also becomes unrepresentative for a dataset if $s_D^* - s_D^b$ is very small (because we divide by $s_D^* - s_D^b$).

If the edge case happens, we set the score of all methods worse than the baseline to $-10$, following a penalization-like approach (*e.g.*, PAR10 from Algorithm Selection (Lindauer et al., 2019)). Methods for which $|s_D - s_D^b| <= \Delta$ holds are assigned a score of $-1$.

Table 2: Supplementary information for all datasets and generated base models.

| Dataset Name | OpenML Task ID | #Instances | #Features | #Classes | Memory (GB) | Mean #Base Models | | | | Mean #Distinct Algorithms | | | |
|---|---|---|---|---|---|---|---|---|---|---|---|---|---|
| | | | | | | $BA_{TopN}$ | $BA_{SiloTopN}$ | $R_{TopN}$ | $R_{SiloTopN}$ | $BA_{TopN}$ | $BA_{SiloTopN}$ | $R_{TopN}$ | $R_{SiloTopN}$ |
| yeast | 2073 | 1484 | 9 | 10 | 32 | 50.0 | 50.0 | 50.0 | 50.0 | 1.7 | 16.0 | 2.1 | 16.0 |
| KDDCup09_appetency | 3945 | 50000 | 231 | 2 | 32 | 50.0 | 50.0 | 50.0 | 50.0 | 1.5 | 16.0 | 1.2 | 15.8 |
| covertype | 7593 | 581012 | 55 | 7 | 64 | 49.9 | 49.9 | 50.0 | 50.0 | 10.0 | 12.6 | 7.9 | 12.2 |
| amazon-commerce-reviews | 10090 | 1500 | 10001 | 50 | 32 | 50.0 | 50.0 | 50.0 | 50.0 | 1.4 | 15.8 | 1.8 | 15.9 |
| Australian | 146818 | 690 | 15 | 2 | 32 | 50.0 | 50.0 | 50.0 | 50.0 | 2.8 | 16.0 | 2.3 | 16.0 |
| wilt | 146820 | 4839 | 6 | 2 | 32 | 50.0 | 50.0 | 50.0 | 50.0 | 1.3 | 16.0 | 1.6 | 16.0 |
| numerai28.6 | 167120 | 96320 | 22 | 2 | 32 | 50.0 | 50.0 | 50.0 | 50.0 | 5.3 | 15.8 | 5.0 | 15.7 |
| phoneme | 168350 | 5404 | 6 | 2 | 32 | 50.0 | 50.0 | 50.0 | 50.0 | 1.3 | 16.0 | 1.3 | 16.0 |
| credit-g | 168757 | 1000 | 21 | 2 | 32 | 50.0 | 50.0 | 50.0 | 50.0 | 1.5 | 16.0 | 2.6 | 16.0 |
| steel-plates-fault | 168784 | 1941 | 28 | 7 | 32 | 50.0 | 50.0 | 50.0 | 50.0 | 1.4 | 16.0 | 1.1 | 16.0 |
| APSFailure | 168868 | 76000 | 171 | 2 | 32 | 50.0 | 50.0 | 50.0 | 50.0 | 2.5 | 16.0 | 2.0 | 16.0 |
| dilbert | 168909 | 10000 | 2001 | 5 | 32 | 50.0 | 50.0 | 50.0 | 50.0 | 1.5 | 15.7 | 1.0 | 15.7 |
| fabert | 168910 | 8237 | 801 | 7 | 32 | 50.0 | 50.0 | 50.0 | 50.0 | 3.4 | 16.0 | 2.6 | 16.0 |
| jasmine | 168911 | 2984 | 145 | 2 | 32 | 50.0 | 50.0 | 50.0 | 50.0 | 3.0 | 15.8 | 1.9 | 15.9 |
| airlines | 189354 | 539383 | 8 | 2 | 64 | 50.0 | 50.0 | 50.0 | 50.0 | 5.8 | 14.5 | 5.9 | 14.4 |
| dionis | 189355 | 416188 | 61 | 355 | 128 | 20.8 | 20.8 | 28.2 | 28.2 | 7.4 | 7.4 | 7.9 | 7.9 |
| albert | 189356 | 425240 | 79 | 2 | 64 | 50.0 | 50.0 | 50.0 | 50.0 | 8.1 | 13.6 | 8.6 | 12.5 |
| gina | 189922 | 3153 | 971 | 2 | 32 | 50.0 | 50.0 | 50.0 | 50.0 | 1.0 | 16.0 | 1.0 | 16.0 |
| ozone-level-8hr | 190137 | 2534 | 73 | 2 | 32 | 50.0 | 50.0 | 50.0 | 50.0 | 1.0 | 16.0 | 1.8 | 15.9 |
| vehicle | 190146 | 846 | 19 | 4 | 32 | 50.0 | 50.0 | 50.0 | 50.0 | 1.6 | 16.0 | 1.6 | 16.0 |
| madeline | 190392 | 3140 | 260 | 2 | 32 | 50.0 | 50.0 | 50.0 | 50.0 | 1.2 | 16.0 | 1.3 | 16.0 |
| philippine | 190410 | 5832 | 309 | 2 | 32 | 50.0 | 50.0 | 50.0 | 50.0 | 1.0 | 16.0 | 1.2 | 16.0 |
| ada | 190411 | 4147 | 49 | 2 | 32 | 50.0 | 50.0 | 50.0 | 50.0 | 1.9 | 16.0 | 1.6 | 16.0 |
| arcene | 190412 | 100 | 10001 | 2 | 32 | 50.0 | 50.0 | 50.0 | 50.0 | 1.3 | 16.0 | 1.2 | 16.0 |
| jannis | 211979 | 83733 | 55 | 4 | 32 | 50.0 | 50.0 | 50.0 | 50.0 | 1.0 | 15.4 | 3.1 | 15.1 |
| Diabetes130US | 211986 | 101766 | 50 | 3 | 32 | 50.0 | 50.0 | 50.0 | 50.0 | 1.0 | 15.6 | 1.0 | 15.7 |
| micro-mass | 359953 | 571 | 1301 | 20 | 32 | 50.0 | 50.0 | 50.0 | 50.0 | 1.1 | 16.0 | 1.5 | 16.0 |
| eucalyptus | 359954 | 736 | 20 | 5 | 32 | 50.0 | 50.0 | 50.0 | 50.0 | 1.3 | 16.0 | 2.6 | 15.9 |
| blood-transfusion-service-center | 359955 | 748 | 5 | 2 | 32 | 50.0 | 50.0 | 50.0 | 50.0 | 1.9 | 16.0 | 4.1 | 16.0 |
| qsar-biodeg | 359956 | 1055 | 42 | 2 | 32 | 50.0 | 50.0 | 50.0 | 50.0 | 2.4 | 16.0 | 2.3 | 16.0 |
| cnae-9 | 359957 | 1080 | 857 | 9 | 32 | 50.0 | 50.0 | 50.0 | 50.0 | 1.6 | 16.0 | 1.2 | 16.0 |
| pc4 | 359958 | 1458 | 38 | 2 | 32 | 50.0 | 50.0 | 50.0 | 50.0 | 1.4 | 16.0 | 2.1 | 16.0 |
| cmc | 359959 | 1473 | 10 | 3 | 32 | 50.0 | 50.0 | 50.0 | 50.0 | 1.9 | 16.0 | 1.6 | 16.0 |
| car | 359960 | 1728 | 7 | 4 | 32 | 50.0 | 50.0 | 50.0 | 50.0 | 1.1 | 16.0 | 1.0 | 16.0 |
| mfeat-factors | 359961 | 2000 | 217 | 10 | 32 | 50.0 | 50.0 | 50.0 | 50.0 | 1.1 | 16.0 | 3.5 | 16.0 |
| kc1 | 359962 | 2109 | 22 | 2 | 32 | 50.0 | 50.0 | 50.0 | 50.0 | 2.2 | 16.0 | 2.2 | 16.0 |
| segment | 359963 | 2310 | 20 | 7 | 32 | 50.0 | 50.0 | 50.0 | 50.0 | 1.4 | 16.0 | 1.4 | 16.0 |
| dna | 359964 | 3186 | 181 | 3 | 32 | 50.0 | 50.0 | 50.0 | 50.0 | 1.2 | 16.0 | 1.1 | 15.8 |
| kr-vs-kp | 359965 | 3196 | 37 | 2 | 32 | 50.0 | 50.0 | 50.0 | 50.0 | 1.8 | 16.0 | 1.1 | 15.8 |
| Internet-Advertisements | 359966 | 3279 | 1559 | 2 | 32 | 50.0 | 50.0 | 50.0 | 50.0 | 1.8 | 16.0 | 2.6 | 16.0 |
| Bioresponse | 359967 | 3751 | 1777 | 2 | 32 | 50.0 | 50.0 | 50.0 | 50.0 | 3.6 | 15.9 | 2.6 | 15.8 |
| churn | 359968 | 5000 | 21 | 2 | 32 | 50.0 | 50.0 | 50.0 | 50.0 | 1.3 | 16.0 | 1.2 | 16.0 |
| first-order-theorem-proving | 359969 | 6118 | 52 | 6 | 32 | 50.0 | 50.0 | 50.0 | 50.0 | 2.8 | 15.8 | 2.6 | 15.7 |
| GesturePhaseSegmentationProcessed | 359970 | 9873 | 33 | 5 | 32 | 50.0 | 50.0 | 50.0 | 50.0 | 1.1 | 15.8 | 2.0 | 15.9 |
| PhishingWebsites | 359971 | 11055 | 31 | 2 | 32 | 50.0 | 50.0 | 50.0 | 50.0 | 1.1 | 15.8 | 1.1 | 15.6 |
| sylvine | 359972 | 5124 | 21 | 2 | 32 | 50.0 | 50.0 | 50.0 | 50.0 | 1.3 | 16.0 | 1.1 | 16.0 |
| christine | 359973 | 5418 | 1637 | 2 | 32 | 50.0 | 50.0 | 50.0 | 50.0 | 3.5 | 15.8 | 2.9 | 15.9 |
| wine-quality-white | 359974 | 4898 | 12 | 7 | 32 | 50.0 | 50.0 | 50.0 | 50.0 | 1.0 | 16.0 | 1.7 | 16.0 |
| Satellite | 359975 | 5100 | 37 | 2 | 32 | 50.0 | 50.0 | 50.0 | 50.0 | 1.2 | 16.0 | 1.2 | 16.0 |
| Fashion-MNIST | 359976 | 70000 | 785 | 10 | 64 | 50.0 | 50.0 | 50.0 | 50.0 | 12.3 | 13.4 | 6.1 | 13.0 |
| connect-4 | 359977 | 67557 | 43 | 3 | 32 | 50.0 | 50.0 | 50.0 | 50.0 | 1.0 | 16.0 | 1.0 | 15.7 |
| Amazon_employee_access | 359979 | 32769 | 10 | 2 | 32 | 50.0 | 50.0 | 50.0 | 50.0 | 2.5 | 15.9 | 1.7 | 16.0 |
| nomao | 359980 | 34465 | 119 | 2 | 32 | 50.0 | 50.0 | 50.0 | 50.0 | 1.0 | 15.9 | 1.0 | 15.9 |
| jungle_chess_2pcs_raw_endgame_complete | 359981 | 44819 | 7 | 3 | 32 | 50.0 | 50.0 | 50.0 | 50.0 | 1.0 | 16.0 | 1.0 | 16.0 |
| bank-marketing | 359982 | 45211 | 17 | 2 | 32 | 50.0 | 50.0 | 50.0 | 50.0 | 1.0 | 15.8 | 1.2 | 15.8 |
| adult | 359983 | 48842 | 15 | 2 | 32 | 50.0 | 50.0 | 50.0 | 50.0 | 1.3 | 16.0 | 1.0 | 16.0 |
| helena | 359984 | 65196 | 28 | 100 | 32 | 50.0 | 50.0 | 50.0 | 50.0 | 2.8 | 15.8 | 3.6 | 15.2 |
| volkert | 359985 | 58310 | 181 | 10 | 32 | 50.0 | 50.0 | 50.0 | 50.0 | 2.1 | 15.5 | 1.1 | 15.5 |
| robert | 359986 | 10000 | 7201 | 10 | 64 | 50.0 | 50.0 | 50.0 | 50.0 | 11.2 | 14.3 | 7.7 | 13.7 |
| shuttle | 359987 | 58000 | 10 | 7 | 32 | 50.0 | 50.0 | 50.0 | 50.0 | 1.2 | 16.0 | 2.1 | 16.0 |
| guillermo | 359988 | 20000 | 4297 | 2 | 32 | 50.0 | 50.0 | 50.0 | 50.0 | 3.1 | 15.5 | 2.9 | 15.6 |
| riccardo | 359989 | 20000 | 4297 | 2 | 64 | 50.0 | 50.0 | 50.0 | 50.0 | 1.5 | 15.6 | 1.1 | 15.5 |
| MiniBooNE | 359990 | 130064 | 51 | 2 | 32 | 50.0 | 50.0 | 50.0 | 50.0 | 1.4 | 15.7 | 1.1 | 15.4 |
| kick | 359991 | 72983 | 33 | 2 | 32 | 50.0 | 50.0 | 50.0 | 50.0 | 2.0 | 15.9 | 1.2 | 15.8 |
| Click_prediction_small | 359992 | 39948 | 12 | 2 | 32 | 50.0 | 50.0 | 50.0 | 50.0 | 1.0 | 16.0 | 1.0 | 15.9 |
| okcupid-stem | 359993 | 50789 | 20 | 3 | 32 | 50.0 | 50.0 | 50.0 | 50.0 | 1.1 | 15.9 | 1.0 | 15.5 |
| sf-police-incidents | 359994 | 2215023 | 9 | 2 | 64 | 49.5 | 49.5 | 50.0 | 50.0 | 10.5 | 12.9 | 6.1 | 13.1 |
| KDDCup99 | 360112 | 4898431 | 42 | 23 | 128 | 13.5 | 13.5 | 29.2 | 29.2 | 5.1 | 5.1 | 9.2 | 9.2 |
| porto-seguro | 360113 | 595212 | 58 | 2 | 64 | 50.0 | 50.0 | 50.0 | 50.0 | 10.6 | 14.4 | 10.8 | 14.5 |
| Higgs | 360114 | 1000000 | 29 | 2 | 64 | 47.9 | 47.9 | 48.8 | 48.8 | 9.8 | 13.2 | 11.2 | 11.8 |
| KDDCup09-Upselling | 360975 | 50000 | 14892 | 2 | 128 | 50.0 | 50.0 | 49.0 | 49.0 | 4.9 | 13.4 | 6.0 | 11.7 |

Table 3: Q(D)O-ES Configuration Space.

| Hyperparameter | Values |
|---|---|
| Preprocessing | SiloTopN, TopN |
| Batch Size | 20, 40 |
| Archive Size | 16, 49 |
| Archive Initialisation | L1 Ensembles, L2 of Single Best, Random L2 |
| Sampling Method | Deterministic, Tournament, Dynamic |
| Crossover | two-point crossover, average crossover, no crossover |

## F  Supplements for the Results

### F.1  Estimated Lower Bound of Wall-Clock Time for the Experiments

We estimated the lower bound for the wall-clock time for our experiments to be approximately 3.91 years. We did not capture the CPU time while running our experiments and cannot calculate this in retrospect due to parallelization. We note, that the wall-clock time is an absolute lower bound for the CPU time. Moreover, for our estimate of wall-clock time, we exclude the overhead that parallelization and containerization induced, because we did not time this overhead. Nevertheless, we believe this to be a non-marginal overhead, considering the considerable number of ensemble runs and our experiences of running the experiments.

To compute the estimate, we first consider the wall-clock time it took to generate the base models with Auto-Sklearn. Therefore, we multiply the budget of 4 hours that we gave to Auto-Sklearn with 8 cores: 4 hours × 10 folds × 71 datasets × 2 metrics; amounting to ~0.65 years.

Next, we consider the wall-clock time it took to run all configurations of all ensemble methods. Therefore, we use the time it took to fit and predict with the ensemble methods. We captured this while evaluating the ensemble methods (the related data can be found in our code repository). Note, that the fit and predict time measurement excludes any time spent related to the base models, as the evaluation only works on predictions of the base models stored after their generation – this excludes loading the predictions or the datasets. Likewise, the measured time excludes the time it took to score ensemble methods and store the results related to the evaluation. Lastly, the ensemble method had 8 cores available for fitting and predicting. Given these constraints, we estimate a lower bound by summing the fit and predict time of all configurations across all datasets, folds, configurations, and both metrics; amounting to ~3.26 years.

In conclusion, we estimated the lower bound for the wall-clock time for our experiments to be approximately 3.91 years without parallelization across datasets, folds, configurations, or metrics.

### F.2  Additional Results on Validation Data

For a visualisation of the distribution of the relative performance of the ensemble methods on validation data, see Figure 4 for boxplots of the normalised improvement. We observe that the normalised improvement for QDO-ES is substantially larger than for GES and QO-ES.

### F.3  Ablation Study: Q(D)O-ES Hyperparameter Importance

We perform an ablation study of the components of Q(D)O-ES by determining the importance of hyperparameters on the final performance.

We analyse the importance of hyperparameters using fANOVA (Hutter et al., 2014). Therefore, we compute the importance of the hyperparameters defined in Table 3 for each dataset for QO-ES and QDO-ES. Table 4 shows the mean importance (higher is more important) over all datasets split by metric and classification task for QDO-ES and QO-ES.

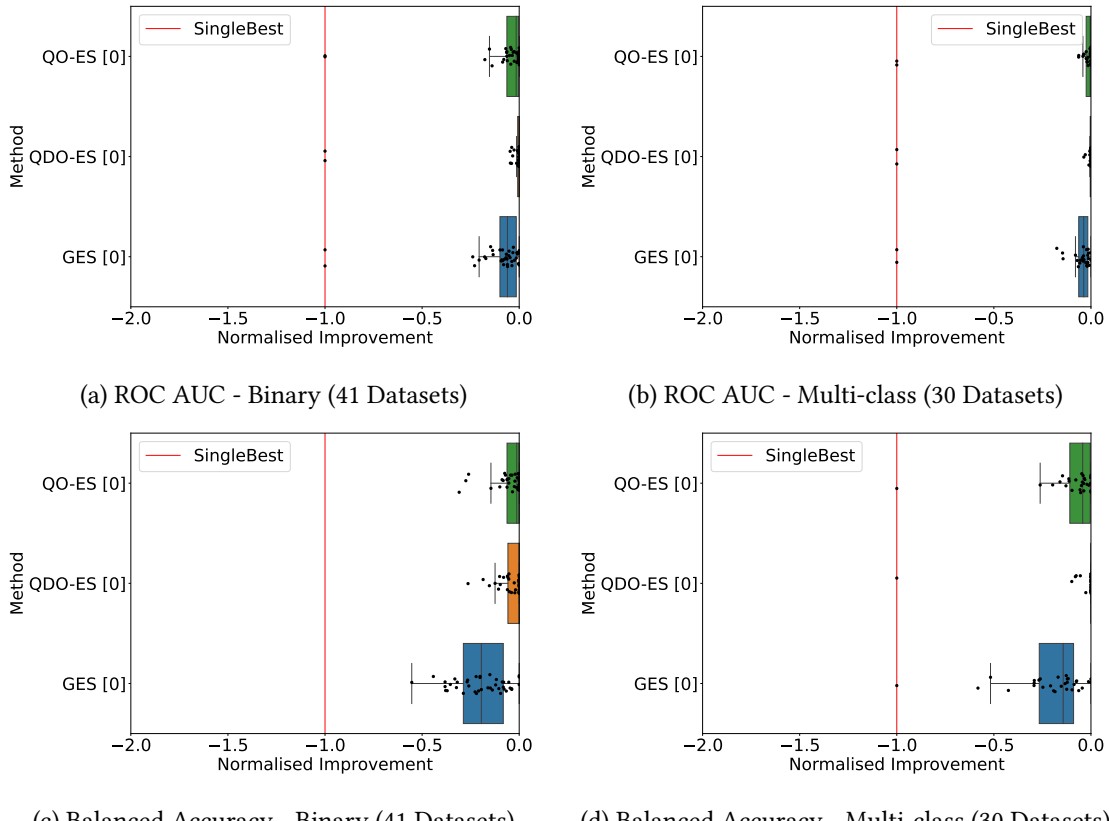

(a) ROC AUC - Binary (41 Datasets)

(b) ROC AUC - Multi-class (30 Datasets)

(c) Balanced Accuracy - Binary (41 Datasets)

(d) Balanced Accuracy - Multi-class (30 Datasets)

Figure 4: **Normalised Improvement Boxplots for Validation Scores**: Higher normalised improvement is better. Each black point represents the improvement for one dataset.

Table 4: Mean hyperparameter importance computed with fANOVA for QO-ES and QDO-ES (higher means more important).

| | QDO-ES | | | | QO-ES | | | |
| | Balanced Accuracy | | ROC AUC | | Balanced Accuracy | | ROC AUC | |
| Hyperparameter | Binary | Multi-class | Binary | Multi-class | Binary | Multi-class | Binary | Multi-class |
|---|---|---|---|---|---|---|---|---|
| Preprocessing | 0.57 | 0.52 | 0.68 | 0.51 | 0.60 | 0.75 | 0.69 | 0.73 |
| Batch Size | 0.03 | 0.03 | 0.02 | 0.03 | 0.03 | 0.01 | 0.02 | 0.02 |
| Archive Size | 0.02 | 0.02 | 0.01 | 0.03 | 0.02 | 0.01 | 0.01 | 0.01 |
| Archive Initialisation | 0.15 | 0.16 | 0.11 | 0.21 | 0.14 | 0.12 | 0.10 | 0.10 |
| Sampling Method | 0.15 | 0.18 | 0.12 | 0.16 | 0.14 | 0.06 | 0.11 | 0.07 |
| Crossover | 0.08 | 0.09 | 0.06 | 0.07 | 0.07 | 0.06 | 0.06 | 0.06 |

We observe that "Preprocessing", that is, the pruning method before we perform *post hoc* ensembling, is by far the most important hyperparameter across all scenarios for both QDO-ES and QO-ES. "Archive Initialisation" and "Sampling Method" follow with a similar distance of importance to "Preprocessing" and seem to be important as well. These are followed by "Crossover" with a lower relative importance, while varying "Batch Size" or "Archive Size" do not seem to have any meaningful impact on the final performance.

Table 5: Median normalised improvement per value of a hyperparameter for QO-ES and QDO-ES (higher is better). A performance of $-1$ is equal to the performance of the SingleBest model, and a performance of 0 is equal to the performance of the best-observed configuration per dataset.

| | | QDO-ES | | | | QO-ES | | | |
| | | Balanced Accuracy | | ROC AUC | | Balanced Accuracy | | ROC AUC | |
| Hyperparameter | Value | Binary | Multi-class | Binary | Multi-class | Binary | Multi-class | Binary | Multi-class |
|---|---|---|---|---|---|---|---|---|---|
| Preprocessing | SiloTopN | -0.65 | -0.42 | -0.27 | -0.16 | -0.63 | -0.46 | -0.26 | -0.15 |
| | TopN | -0.67 | -0.63 | -0.33 | -0.39 | -0.65 | -0.63 | -0.29 | -0.36 |
| Batch Size | 20 | -0.68 | -0.51 | -0.3 | -0.24 | -0.66 | -0.49 | -0.29 | -0.22 |
| | 40 | -0.67 | -0.48 | -0.3 | -0.25 | -0.65 | -0.5 | -0.31 | -0.23 |
| Archive Size | 16 | -0.66 | -0.49 | -0.3 | -0.24 | -0.66 | -0.51 | -0.3 | -0.22 |
| | 49 | -0.69 | -0.5 | -0.3 | -0.25 | -0.65 | -0.48 | -0.3 | -0.22 |
| Archive Initialisation | L1 Ensembles | -0.7 | -0.48 | -0.3 | -0.24 | -0.62 | -0.46 | -0.3 | -0.23 |
| | L2 of Single Best | -0.65 | -0.51 | -0.32 | -0.25 | -0.65 | -0.53 | -0.31 | -0.23 |
| | Random L2 | -0.48 | -0.48 | -0.24 | -0.25 | -0.66 | -0.48 | -0.29 | -0.23 |
| Sampling Method | Deterministic | -0.74 | -0.54 | -0.31 | -0.25 | -0.68 | -0.53 | -0.31 | -0.24 |
| | Tournament | -0.63 | -0.47 | -0.3 | -0.24 | -0.64 | -0.48 | -0.29 | -0.22 |
| | Dynamic | -0.64 | -0.48 | -0.3 | -0.25 | -0.62 | -0.45 | -0.3 | -0.22 |
| Crossover | Two-Point | -0.68 | -0.47 | -0.31 | -0.25 | -0.65 | -0.49 | -0.3 | -0.22 |
| | Average | -0.64 | -0.49 | -0.3 | -0.24 | -0.65 | -0.49 | -0.28 | -0.22 |
| | No | -0.68 | -0.53 | -0.3 | -0.25 | -0.67 | -0.5 | -0.31 | -0.23 |

## F.4 Ablation Study: Performance of Hyperparameter Values for Q(D)O-ES

We perform an ablation study of the configurations of the components of Q(D)O-ES by determining the impact of individual hyperparameter values on the final performance.

To assess this performance of individual hyperparameter values, we compute the aggregated normalised improvement per hyperparameter value. That is, we first compute the normalised improvement for all configurations per dataset. Then, we split all configurations based on hyperparameter values and compute the average across all configurations per split; resulting in an average score per dataset for a specific hyperparameter value. Finally, following similar arguments as in Appendix E.3, we compute the median across all datasets per scenario (i.e., the combination of metric and task type). This value now shows the median performance attributed to the hyperparameter value and allows us to interpret how well certain hyperparameter values perform aggregated across datasets. See Table 5 for an overview of the median normalised improvement for all hyperparameters.

We observe that the largest difference in performance between hyperparameter values occurs for "Preprocessing"; mirroring the importance of these hyperparameters shown in Table 4. Likewise, the largest difference in performance can be seen for "Preprocessing" in multi-class classification. Likely a result of overfitting, as overfitting is potentially less severe for multi-class classification (Feldman et al., 2019).

## G  Essential Python Frameworks for the Implementation and Experiments

The following frameworks were essential for our implementation and experiments:

- Auto-Sklearn 1.0 (Feurer et al., 2015), Version: 0.14.7, BSD-3-Clause License; We used Auto-Sklearn to generate base models and took the initial code for greedy ensemble selection from Auto-Sklearn's implementation.

- pyribs (Tjanaka et al., 2021), Version 0.4.0, MIT License; We used pyribs' framework to implement population-based search and the archives.

- Assembled (Purucker and Beel, 2022), Version 0.0.4, MIT License; We used Assembled to store the base models generated with Auto-Sklearn and to run our ensemble-related experiments.

## H  DOIs for Data and Code

The following assets were newly created as part of our experiments:

- The code for our experiments: `https://doi.org/10.6084/m9.figshare.23613624`.

- The prediction data of base models collected by running Auto-Sklearn 1.0 on the classification datasets from the AutoML benchmark: `https://doi.org/10.6084/m9.figshare.23613627`.

## I  Test and Validation Performance Per Dataset Per Scenario

The following tables provide the exact performance of each ensembling method per dataset. The tables are split per scenario (i.e., the combinations of metric and task type).

Table 6: **Test ROC AUC - Binary**: The mean and standard deviation of the test score over all folds for each method. The best methods per dataset are shown in bold. All methods close to the best method are considered best (using NumPy's default *isclose* function).

| Dataset | GES | QDO-ES | QO-ES | SingleBest |
|---|---|---|---|---|
| APSFailure | 0.9917 (± 0.0036) | 0.9918 (± 0.0041) | **0.9921 (± 0.0036)** | 0.9912 (± 0.0029) |
| Amazon_employee_access | 0.8681 (± 0.015) | 0.8684 (± 0.0134) | **0.8684 (± 0.014)** | 0.8616 (± 0.017) |
| Australian | 0.9317 (± 0.0198) | 0.9329 (± 0.0236) | **0.9349 (± 0.0202)** | 0.927 (± 0.0255) |
| Bioresponse | 0.8724 (± 0.0156) | 0.8721 (± 0.0169) | **0.8731 (± 0.0163)** | 0.8675 (± 0.0196) |
| Click_prediction_small | 0.7004 (± 0.0142) | **0.7009 (± 0.0139)** | 0.7009 (± 0.0137) | 0.6964 (± 0.0152) |
| Higgs | 0.841 (± 0.0017) | 0.841 (± 0.0017) | **0.841 (± 0.0017)** | 0.8328 (± 0.002) |
| Internet-Advertisements | **0.987 (± 0.0076)** | 0.9863 (± 0.0063) | 0.9859 (± 0.0077) | 0.9759 (± 0.0196) |
| KDDCup09-Upselling | 0.9077 (± 0.009) | 0.908 (± 0.0085) | **0.9082 (± 0.0085)** | 0.9058 (± 0.01) |
| KDDCup09_appetency | 0.836 (± 0.0114) | 0.8374 (± 0.0126) | **0.8385 (± 0.0125)** | 0.8292 (± 0.0148) |
| MiniBooNE | **0.9874 (± 0.001)** | 0.9874 (± 0.0011) | 0.9874 (± 0.0011) | 0.9871 (± 0.001) |
| PhishingWebsites | **0.9968 (± 0.0008)** | 0.9967 (± 0.001) | 0.9966 (± 0.0012) | 0.9963 (± 0.0011) |
| Satellite | 0.9836 (± 0.0287) | 0.9871 (± 0.019) | **0.9886 (± 0.014)** | 0.9788 (± 0.03) |
| ada | 0.9169 (± 0.019) | **0.9172 (± 0.0201)** | 0.917 (± 0.0199) | 0.9155 (± 0.0176) |
| adult | **0.9294 (± 0.0046)** | 0.9294 (± 0.0045) | 0.9294 (± 0.0046) | 0.9286 (± 0.0048) |
| airlines | 0.727 (± 0.003) | 0.7271 (± 0.0031) | **0.7271 (± 0.003)** | 0.7208 (± 0.0033) |
| albert | **0.7623 (± 0.0026)** | 0.7623 (± 0.0027) | 0.7622 (± 0.0025) | 0.7592 (± 0.0038) |
| arcene | 0.8933 (± 0.0981) | 0.897 (± 0.0829) | 0.8977 (± 0.103) | **0.8977 (± 0.1084)** |
| bank-marketing | **0.9388 (± 0.0068)** | 0.9387 (± 0.0067) | **0.9388 (± 0.0067)** | 0.9369 (± 0.0074) |
| blood-transfusion-service-center | **0.7506 (± 0.0459)** | 0.7467 (± 0.0606) | 0.7407 (± 0.0591) | 0.7383 (± 0.0386) |
| christine | 0.831 (± 0.0146) | 0.8314 (± 0.0144) | **0.8317 (± 0.0143)** | 0.8264 (± 0.0138) |
| churn | 0.9193 (± 0.0266) | 0.9212 (± 0.0224) | **0.9215 (± 0.0238)** | 0.9215 (± 0.0245) |
| credit-g | 0.7834 (± 0.0388) | 0.7849 (± 0.0374) | **0.7863 (± 0.0333)** | 0.766 (± 0.0383) |
| gina | 0.9936 (± 0.0039) | 0.9936 (± 0.004) | **0.9937 (± 0.004)** | 0.9917 (± 0.0049) |
| guillermo | 0.9151 (± 0.0091) | **0.9158 (± 0.0087)** | 0.9157 (± 0.0086) | 0.9117 (± 0.0079) |
| jasmine | 0.8825 (± 0.0184) | **0.8831 (± 0.019)** | 0.8824 (± 0.0182) | 0.8816 (± 0.0216) |
| kc1 | 0.8364 (± 0.0374) | 0.8384 (± 0.0355) | **0.8388 (± 0.0347)** | 0.8362 (± 0.0319) |
| kick | 0.7869 (± 0.0077) | 0.787 (± 0.0076) | **0.7871 (± 0.0076)** | 0.7808 (± 0.0059) |
| kr-vs-kp | **0.9995 (± 0.0007)** | 0.9995 (± 0.0008) | 0.9994 (± 0.0008) | 0.9994 (± 0.0007) |
| madeline | 0.969 (± 0.0061) | 0.9689 (± 0.0053) | **0.9692 (± 0.0055)** | 0.9671 (± 0.0051) |
| nomao | 0.9961 (± 0.0009) | 0.9961 (± 0.001) | **0.9961 (± 0.0009)** | 0.9959 (± 0.001) |
| numerai28.6 | **0.5308 (± 0.0036)** | 0.5306 (± 0.0037) | 0.5306 (± 0.0034) | 0.5294 (± 0.0043) |
| ozone-level-8hr | 0.9178 (± 0.0265) | **0.9204 (± 0.029)** | 0.9184 (± 0.0275) | 0.912 (± 0.0257) |
| pc4 | **0.9448 (± 0.0213)** | 0.9438 (± 0.0223) | 0.9435 (± 0.0203) | 0.9331 (± 0.0308) |
| philippine | 0.9149 (± 0.0101) | **0.915 (± 0.0103)** | 0.914 (± 0.0117) | 0.9134 (± 0.0114) |
| phoneme | 0.964 (± 0.0069) | **0.9643 (± 0.0064)** | 0.9643 (± 0.0064) | 0.9637 (± 0.007) |
| porto-seguro | 0.6401 (± 0.0035) | **0.6408 (± 0.0041)** | 0.6408 (± 0.004) | 0.6356 (± 0.0045) |
| qsar-biodeg | 0.928 (± 0.0297) | **0.93 (± 0.0294)** | 0.9288 (± 0.0288) | 0.9184 (± 0.0323) |
| riccardo | 0.9997 (± 0.0003) | **0.9998 (± 0.0002)** | 0.9997 (± 0.0002) | 0.9997 (± 0.0002) |
| sf-police-incidents | 0.7084 (± 0.0037) | 0.7091 (± 0.0033) | **0.7092 (± 0.0034)** | 0.6924 (± 0.0072) |
| sylvine | **0.9927 (± 0.0034)** | 0.9923 (± 0.0033) | 0.9925 (± 0.0033) | 0.9915 (± 0.0037) |
| wilt | 0.9953 (± 0.0047) | 0.9953 (± 0.0044) | **0.9957 (± 0.0034)** | 0.9949 (± 0.0045) |

Table 7: **Test ROC AUC - Multi-class**: The mean and standard deviation of the test score over all folds for each method. The best methods per dataset are shown in bold. All methods close to the best method are considered best (using NumPy's default *isclose* function).

| Dataset | GES | QDO-ES | QO-ES | SingleBest |
|---|---|---|---|---|
| Diabetes130US | 0.7143 (± 0.0064) | 0.7142 (± 0.0064) | **0.7144 (± 0.0067)** | 0.7099 (± 0.0051) |
| Fashion-MNIST | 0.9945 (± 0.0005) | **0.9946 (± 0.0005)** | 0.9945 (± 0.0005) | 0.9942 (± 0.0005) |
| GesturePhaseSegmentationProcessed | 0.9137 (± 0.0086) | 0.9137 (± 0.0083) | **0.9143 (± 0.0085)** | 0.9039 (± 0.0055) |
| KDDCup99 | 0.9987 (± 0.0042) | 0.9992 (± 0.0018) | 0.9991 (± 0.0018) | **0.9997 (± 0.0007)** |
| amazon-commerce-reviews | 0.9946 (± 0.0032) | 0.9945 (± 0.0034) | **0.9948 (± 0.0029)** | 0.9859 (± 0.0052) |
| car | 0.9998 (± 0.0006) | **1.0 (± 0.0001)** | **1.0 (± 0.0001)** | **1.0 (± 0.0001)** |
| cmc | 0.7475 (± 0.0329) | 0.7468 (± 0.0315) | **0.7477 (± 0.0328)** | 0.7391 (± 0.0298) |
| cnae-9 | 0.9977 (± 0.0019) | 0.9978 (± 0.0015) | **0.998 (± 0.0011)** | 0.9952 (± 0.0037) |
| connect-4 | 0.9405 (± 0.0039) | 0.9408 (± 0.0038) | **0.9408 (± 0.0039)** | 0.9371 (± 0.0026) |
| covertype | **0.9991 (± 0.0001)** | **0.9991 (± 0.0001)** | **0.9991 (± 0.0001)** | 0.9988 (± 0.0001) |
| dilbert | 0.9999 (± 0.0001) | **0.9999 (± 0.0)** | **0.9999 (± 0.0)** | 0.9998 (± 0.0001) |
| dionis | **0.9993 (± 0.0001)** | 0.9993 (± 0.0001) | 0.9993 (± 0.0001) | 0.9987 (± 0.0003) |
| dna | 0.9947 (± 0.0033) | **0.995 (± 0.0032)** | 0.995 (± 0.0032) | 0.9941 (± 0.0036) |
| eucalyptus | **0.9222 (± 0.016)** | 0.9205 (± 0.017) | 0.9212 (± 0.0148) | 0.9162 (± 0.0172) |
| fabert | **0.9428 (± 0.0034)** | 0.9428 (± 0.0033) | 0.9427 (± 0.0033) | 0.9334 (± 0.0044) |
| first-order-theorem-proving | 0.8398 (± 0.0118) | 0.8398 (± 0.0113) | **0.84 (± 0.0112)** | 0.8359 (± 0.0127) |
| helena | 0.8917 (± 0.0025) | 0.8919 (± 0.0025) | **0.8919 (± 0.0025)** | 0.8778 (± 0.0029) |
| jannis | 0.8853 (± 0.0034) | **0.8856 (± 0.0034)** | **0.8856 (± 0.0033)** | 0.8802 (± 0.0035) |
| jungle_chess_2pcs_raw_endgame_complete | **0.9927 (± 0.0017)** | 0.9926 (± 0.0016) | 0.9927 (± 0.0016) | 0.9915 (± 0.0022) |
| mfeat-factors | 0.9983 (± 0.0028) | 0.9985 (± 0.0028) | 0.9984 (± 0.0026) | **0.9988 (± 0.0009)** |
| micro-mass | 0.9966 (± 0.0031) | 0.9965 (± 0.0032) | **0.9978 (± 0.0018)** | 0.9939 (± 0.0084) |
| okcupid-stem | 0.8288 (± 0.0056) | **0.829 (± 0.0055)** | 0.8289 (± 0.0054) | 0.8258 (± 0.0054) |
| robert | 0.8982 (± 0.0048) | 0.8984 (± 0.0042) | **0.8987 (± 0.0045)** | 0.8864 (± 0.0066) |
| segment | 0.9956 (± 0.0017) | 0.9953 (± 0.0019) | **0.9958 (± 0.0016)** | 0.9949 (± 0.0021) |
| shuttle | 0.9928 (± 0.0226) | **1.0 (± 0.0)** | **1.0 (± 0.0)** | **1.0 (± 0.0)** |
| steel-plates-fault | 0.9618 (± 0.0074) | **0.9629 (± 0.0067)** | 0.9621 (± 0.0078) | 0.9569 (± 0.0119) |
| vehicle | 0.9679 (± 0.0088) | 0.9671 (± 0.0102) | **0.9682 (± 0.0092)** | 0.9656 (± 0.0089) |
| volkert | 0.9565 (± 0.0023) | **0.957 (± 0.0018)** | 0.9567 (± 0.0021) | 0.9531 (± 0.0022) |
| wine-quality-white | **0.866 (± 0.0329)** | 0.864 (± 0.0284) | 0.8652 (± 0.0331) | 0.8613 (± 0.0394) |
| yeast | **0.8771 (± 0.0386)** | 0.8743 (± 0.0445) | 0.877 (± 0.0371) | 0.8627 (± 0.0365) |

Table 8: **Test Balanced Accuracy - Binary**: The mean and standard deviation of the test score over all folds for each method. The best methods per dataset are shown in bold. All methods close to the best method are considered best (using NumPy's default *isclose* function).

| Dataset | GES | QDO-ES | QO-ES | SingleBest |
|---|---|---|---|---|
| APSFailure | **0.9625 (± 0.0073)** | 0.9608 (± 0.0105) | 0.9594 (± 0.0082) | 0.9539 (± 0.0128) |
| Amazon_employee_access | 0.7914 (± 0.0089) | 0.7913 (± 0.0112) | **0.7923 (± 0.0121)** | 0.7837 (± 0.0142) |
| Australian | 0.8526 (± 0.0331) | 0.8582 (± 0.0285) | **0.8612 (± 0.0307)** | 0.8471 (± 0.0287) |
| Bioresponse | 0.7911 (± 0.0151) | 0.7924 (± 0.0125) | **0.7956 (± 0.0171)** | 0.7946 (± 0.0181) |
| Click_prediction_small | 0.6365 (± 0.0142) | 0.6381 (± 0.0119) | **0.6384 (± 0.0129)** | 0.6377 (± 0.0146) |
| Higgs | 0.7518 (± 0.0016) | 0.7519 (± 0.0017) | **0.752 (± 0.0016)** | 0.7469 (± 0.0027) |
| Internet-Advertisements | 0.9445 (± 0.0203) | **0.9481 (± 0.0215)** | 0.9455 (± 0.024) | 0.9442 (± 0.0225) |
| KDDCup09-Upselling | 0.7985 (± 0.0104) | 0.7993 (± 0.0125) | **0.8002 (± 0.0099)** | 0.8001 (± 0.0105) |
| KDDCup09_appetency | 0.7492 (± 0.0134) | 0.7472 (± 0.022) | **0.7505 (± 0.0163)** | 0.7465 (± 0.0168) |
| MiniBooNE | 0.9462 (± 0.003) | 0.9461 (± 0.0032) | **0.9464 (± 0.0032)** | 0.9443 (± 0.0026) |
| PhishingWebsites | **0.9706 (± 0.0049)** | 0.9695 (± 0.0038) | 0.9696 (± 0.0051) | 0.9696 (± 0.0037) |
| Satellite | 0.8738 (± 0.0728) | 0.8936 (± 0.0832) | 0.8613 (± 0.0597) | **0.925 (± 0.093)** |
| ada | 0.8324 (± 0.0322) | **0.8352 (± 0.0289)** | 0.8339 (± 0.0277) | 0.8316 (± 0.0304) |
| adult | 0.843 (± 0.0073) | 0.8431 (± 0.0079) | **0.8439 (± 0.0079)** | 0.8428 (± 0.0083) |
| airlines | 0.6624 (± 0.0025) | 0.6628 (± 0.0022) | **0.6628 (± 0.0025)** | 0.6583 (± 0.0022) |
| albert | 0.6917 (± 0.0036) | 0.6917 (± 0.0037) | **0.6918 (± 0.0037)** | 0.6892 (± 0.0027) |
| arcene | **0.8208 (± 0.1933)** | 0.7792 (± 0.1827) | 0.7917 (± 0.1784) | 0.7525 (± 0.2149) |
| bank-marketing | 0.874 (± 0.0108) | 0.8739 (± 0.0099) | **0.8746 (± 0.0107)** | 0.8736 (± 0.0098) |
| blood-transfusion-service-center | 0.67 (± 0.0409) | **0.6821 (± 0.0483)** | 0.6746 (± 0.051) | 0.6816 (± 0.055) |
| christine | 0.754 (± 0.015) | **0.7575 (± 0.0156)** | 0.7553 (± 0.0174) | 0.7482 (± 0.0204) |
| churn | 0.9094 (± 0.0143) | 0.9112 (± 0.0191) | **0.912 (± 0.0156)** | 0.9087 (± 0.0161) |
| credit-g | 0.6983 (± 0.0563) | 0.7079 (± 0.0592) | **0.7145 (± 0.035)** | 0.6936 (± 0.0353) |
| gina | **0.9663 (± 0.0106)** | 0.9625 (± 0.0157) | 0.9628 (± 0.0141) | 0.9622 (± 0.0149) |
| guillermo | 0.8366 (± 0.0119) | 0.835 (± 0.0115) | **0.8377 (± 0.0117)** | 0.8341 (± 0.0112) |
| jasmine | 0.8173 (± 0.0205) | 0.8163 (± 0.0169) | **0.8197 (± 0.0189)** | 0.8123 (± 0.0226) |
| kc1 | 0.7363 (± 0.0316) | **0.7543 (± 0.0285)** | 0.7512 (± 0.0345) | 0.7344 (± 0.0285) |
| kick | 0.7003 (± 0.0081) | 0.6993 (± 0.0082) | **0.7024 (± 0.0079)** | 0.698 (± 0.0059) |
| kr-vs-kp | **0.9943 (± 0.0052)** | 0.9933 (± 0.0059) | **0.9943 (± 0.0052)** | 0.993 (± 0.0062) |
| madeline | 0.9108 (± 0.0117) | 0.9073 (± 0.0133) | **0.9169 (± 0.0107)** | 0.9083 (± 0.0171) |
| nomao | 0.9688 (± 0.0029) | 0.9684 (± 0.0026) | **0.9691 (± 0.0024)** | 0.9682 (± 0.0021) |
| numerai28.6 | 0.5204 (± 0.0046) | 0.5204 (± 0.0049) | **0.5215 (± 0.0043)** | 0.5189 (± 0.0047) |
| ozone-level-8hr | 0.8234 (± 0.0353) | 0.8259 (± 0.0383) | 0.8296 (± 0.037) | **0.8337 (± 0.0305)** |
| pc4 | 0.8763 (± 0.0365) | 0.8791 (± 0.051) | **0.883 (± 0.0392)** | 0.8719 (± 0.0406) |
| philippine | **0.8368 (± 0.0108)** | 0.8332 (± 0.0124) | 0.8342 (± 0.0135) | 0.8296 (± 0.0158) |
| phoneme | 0.8902 (± 0.0176) | **0.8914 (± 0.0134)** | 0.8886 (± 0.015) | 0.8893 (± 0.0136) |
| porto-seguro | 0.5991 (± 0.0056) | 0.5991 (± 0.0041) | **0.6 (± 0.0047)** | 0.5989 (± 0.0054) |
| qsar-biodeg | **0.8539 (± 0.039)** | 0.8537 (± 0.0303) | 0.8538 (± 0.041) | 0.8403 (± 0.0271) |
| riccardo | 0.998 (± 0.0007) | **0.9983 (± 0.0007)** | 0.9978 (± 0.0007) | 0.9981 (± 0.0007) |
| sf-police-incidents | 0.6375 (± 0.0043) | **0.6377 (± 0.0032)** | 0.6373 (± 0.0041) | 0.6257 (± 0.0048) |
| sylvine | **0.9567 (± 0.0079)** | **0.9567 (± 0.0068)** | **0.9567 (± 0.0063)** | 0.9561 (± 0.0082) |
| wilt | 0.9676 (± 0.0165) | 0.9644 (± 0.0186) | 0.9622 (± 0.0199) | **0.9771 (± 0.0136)** |

Table 9: **Test Balanced Accuracy - Multi-class:** The mean and standard deviation of the test score over all folds for each method. The best methods per dataset are shown in bold. All methods close to the best method are considered best (using NumPy's default *isclose* function).

| Dataset | GES | QDO-ES | QO-ES | SingleBest |
|---|---|---|---|---|
| Diabetes130US | **0.4996 (± 0.0071)** | 0.4993 (± 0.0065) | 0.4994 (± 0.005) | 0.4964 (± 0.0058) |
| Fashion-MNIST | 0.9124 (± 0.0042) | **0.9129 (± 0.0042)** | 0.9121 (± 0.0038) | 0.9046 (± 0.0062) |
| GesturePhaseSegmentationProcessed | 0.6749 (± 0.0209) | 0.6766 (± 0.0194) | **0.6771 (± 0.0232)** | 0.641 (± 0.0137) |
| KDDCup99 | **0.8113 (± 0.0638)** | 0.8076 (± 0.0677) | 0.8071 (± 0.06) | 0.7985 (± 0.0547) |
| amazon-commerce-reviews | **0.868 (± 0.0279)** | 0.866 (± 0.0284) | 0.862 (± 0.031) | 0.8153 (± 0.0327) |
| car | **0.9932 (± 0.0111)** | 0.9917 (± 0.0113) | 0.9913 (± 0.0125) | 0.9932 (± 0.0066) |
| cmc | 0.5499 (± 0.0441) | 0.5513 (± 0.0408) | **0.5532 (± 0.0328)** | 0.5482 (± 0.0368) |
| cnae-9 | 0.9509 (± 0.0164) | 0.9509 (± 0.0152) | **0.9528 (± 0.0172)** | 0.9454 (± 0.0177) |
| connect-4 | **0.7634 (± 0.009)** | 0.7632 (± 0.0078) | 0.7615 (± 0.0081) | 0.7594 (± 0.0085) |
| covertype | 0.9633 (± 0.0026) | **0.9635 (± 0.003)** | 0.9635 (± 0.0027) | 0.956 (± 0.0031) |
| dilbert | **0.9949 (± 0.0021)** | 0.9945 (± 0.003) | 0.9943 (± 0.0025) | 0.9943 (± 0.0027) |
| dionis | 0.8462 (± 0.017) | **0.8485 (± 0.0163)** | 0.8475 (± 0.0164) | 0.841 (± 0.0107) |
| dna | 0.9619 (± 0.0119) | **0.9655 (± 0.0124)** | 0.9654 (± 0.0117) | 0.959 (± 0.0116) |
| eucalyptus | **0.6463 (± 0.0705)** | 0.6426 (± 0.0658) | 0.62 (± 0.0526) | 0.6264 (± 0.0538) |
| fabert | 0.7115 (± 0.0117) | **0.7148 (± 0.0104)** | 0.7141 (± 0.0104) | 0.6952 (± 0.0146) |
| first-order-theorem-proving | **0.5116 (± 0.0223)** | 0.5107 (± 0.0238) | 0.5069 (± 0.0188) | 0.5028 (± 0.0203) |
| helena | 0.2743 (± 0.0066) | **0.2783 (± 0.005)** | 0.2775 (± 0.0065) | 0.2644 (± 0.0052) |
| jannis | 0.6504 (± 0.0106) | **0.6506 (± 0.0109)** | 0.6504 (± 0.0107) | 0.6484 (± 0.0085) |
| jungle_chess_2pcs_raw_endgame_complete | 0.9261 (± 0.0116) | **0.9268 (± 0.011)** | 0.925 (± 0.0132) | 0.9182 (± 0.0139) |
| mfeat-factors | **0.982 (± 0.0075)** | **0.982 (± 0.0063)** | 0.981 (± 0.0077) | 0.981 (± 0.0077) |
| micro-mass | **0.9137 (± 0.0341)** | 0.906 (± 0.0426) | 0.9088 (± 0.0536) | 0.9108 (± 0.0361) |
| okcupid-stem | **0.6971 (± 0.0083)** | 0.6961 (± 0.0088) | 0.6969 (± 0.0077) | 0.6956 (± 0.0091) |
| robert | 0.5327 (± 0.0115) | **0.5384 (± 0.0125)** | 0.5371 (± 0.0134) | 0.5258 (± 0.0106) |
| segment | 0.9346 (± 0.0183) | 0.9342 (± 0.0201) | **0.9368 (± 0.0179)** | 0.929 (± 0.0218) |
| shuttle | 0.9534 (± 0.0674) | 0.9677 (± 0.0594) | **0.9685 (± 0.0594)** | 0.9677 (± 0.0594) |
| steel-plates-fault | 0.831 (± 0.0254) | 0.8331 (± 0.0234) | **0.8367 (± 0.0208)** | 0.8138 (± 0.033) |
| vehicle | **0.8447 (± 0.0332)** | 0.8343 (± 0.0321) | 0.8366 (± 0.0266) | 0.8352 (± 0.0398) |
| volkert | 0.7087 (± 0.0107) | 0.7085 (± 0.008) | **0.7114 (± 0.0079)** | 0.6699 (± 0.0073) |
| wine-quality-white | **0.4546 (± 0.0463)** | 0.4345 (± 0.0405) | 0.4469 (± 0.0457) | 0.4271 (± 0.0678) |
| yeast | **0.5786 (± 0.0705)** | 0.5728 (± 0.0664) | 0.5657 (± 0.0579) | 0.5631 (± 0.0772) |

Table 10: **Validation ROC AUC - Binary**: The mean and standard deviation of the validation score over all folds for each method. The best methods per dataset are shown in bold. All methods close to the best method are considered best (using NumPy's default *isclose* function).

| Dataset | GES | QDO-ES | QO-ES | SingleBest |
|---|---|---|---|---|
| APSFailure | 0.9952 (± 0.001) | **0.9953 (± 0.001)** | 0.9952 (± 0.0009) | 0.9943 (± 0.001) |
| Amazon_employee_access | 0.8703 (± 0.0072) | **0.8712 (± 0.0076)** | 0.8705 (± 0.0075) | 0.8596 (± 0.0067) |
| Australian | 0.9604 (± 0.0109) | **0.9611 (± 0.0105)** | 0.961 (± 0.0107) | 0.9525 (± 0.0122) |
| Bioresponse | 0.8768 (± 0.0084) | **0.8772 (± 0.0078)** | 0.8771 (± 0.0079) | 0.8701 (± 0.0083) |
| Click_prediction_small | 0.7042 (± 0.0047) | **0.7043 (± 0.0047)** | 0.7042 (± 0.0046) | 0.6999 (± 0.0051) |
| Higgs | 0.8406 (± 0.002) | 0.8406 (± 0.002) | **0.8407 (± 0.0021)** | 0.8325 (± 0.002) |
| Internet-Advertisements | 0.9935 (± 0.003) | **0.9941 (± 0.0019)** | 0.994 (± 0.0019) | 0.9898 (± 0.0029) |
| KDDCup09-Upselling | 0.9124 (± 0.0033) | **0.9131 (± 0.0031)** | 0.9129 (± 0.003) | 0.9092 (± 0.0028) |
| KDDCup09_appetency | 0.8429 (± 0.0106) | **0.8432 (± 0.0105)** | 0.8431 (± 0.0106) | 0.8325 (± 0.0106) |
| MiniBooNE | **0.9875 (± 0.0004)** | 0.9874 (± 0.0004) | 0.9874 (± 0.0005) | 0.987 (± 0.0005) |
| PhishingWebsites | 0.9971 (± 0.0003) | **0.9971 (± 0.0003)** | 0.9971 (± 0.0003) | 0.9968 (± 0.0004) |
| Satellite | 0.9989 (± 0.0012) | **0.9991 (± 0.0011)** | 0.999 (± 0.0013) | 0.9984 (± 0.0013) |
| ada | 0.9235 (± 0.0049) | **0.9239 (± 0.0048)** | 0.9236 (± 0.0048) | 0.9202 (± 0.0058) |
| adult | 0.9301 (± 0.0018) | 0.9301 (± 0.0018) | **0.9301 (± 0.0018)** | 0.9293 (± 0.0018) |
| airlines | 0.7271 (± 0.0015) | 0.7272 (± 0.0016) | **0.7272 (± 0.0015)** | 0.721 (± 0.0019) |
| albert | 0.7621 (± 0.0021) | 0.7621 (± 0.0021) | **0.7621 (± 0.0021)** | 0.7588 (± 0.0035) |
| arcene | 0.9977 (± 0.0049) | **0.9986 (± 0.0043)** | 0.9986 (± 0.0043) | 0.9749 (± 0.0252) |
| bank-marketing | 0.9386 (± 0.0024) | **0.9387 (± 0.0024)** | 0.9387 (± 0.0024) | 0.937 (± 0.0033) |
| blood-transfusion-service-center | 0.7759 (± 0.0098) | **0.7787 (± 0.0117)** | 0.7771 (± 0.0127) | 0.7598 (± 0.0134) |
| christine | 0.8449 (± 0.0079) | **0.845 (± 0.0078)** | 0.8449 (± 0.0079) | 0.835 (± 0.0069) |
| churn | **0.9448 (± 0.0128)** | 0.9446 (± 0.0128) | **0.9448 (± 0.0127)** | 0.9405 (± 0.0134) |
| credit-g | 0.8371 (± 0.0231) | **0.8389 (± 0.022)** | 0.8379 (± 0.0225) | 0.8188 (± 0.0234) |
| gina | **0.9956 (± 0.0015)** | 0.9956 (± 0.0015) | 0.9956 (± 0.0015) | 0.9949 (± 0.0021) |
| guillermo | 0.9162 (± 0.0028) | 0.9168 (± 0.0026) | **0.9169 (± 0.0027)** | 0.9129 (± 0.0017) |
| jasmine | 0.8974 (± 0.012) | 0.8978 (± 0.0122) | **0.8979 (± 0.0121)** | 0.8882 (± 0.0135) |
| kc1 | **0.8598 (± 0.0098)** | 0.8595 (± 0.0095) | 0.8594 (± 0.0092) | 0.8496 (± 0.0103) |
| kick | 0.7923 (± 0.0049) | **0.7924 (± 0.005)** | 0.7923 (± 0.005) | 0.7842 (± 0.0038) |
| kr-vs-kp | 0.9999 (± 0.0002) | **0.9999 (± 0.0001)** | 0.9999 (± 0.0001) | 0.9998 (± 0.0002) |
| madeline | 0.9718 (± 0.0034) | **0.9721 (± 0.0033)** | 0.9719 (± 0.0033) | 0.9681 (± 0.0034) |
| nomao | 0.9964 (± 0.0002) | **0.9964 (± 0.0002)** | 0.9964 (± 0.0002) | 0.9962 (± 0.0002) |
| numerai28.6 | 0.5344 (± 0.0025) | 0.5346 (± 0.0026) | **0.5347 (± 0.0026)** | 0.5312 (± 0.0027) |
| ozone-level-8hr | 0.9554 (± 0.0093) | **0.9571 (± 0.0079)** | 0.956 (± 0.0091) | 0.9486 (± 0.0078) |
| pc4 | 0.9677 (± 0.0081) | **0.9677 (± 0.0075)** | 0.9676 (± 0.0075) | 0.9543 (± 0.0107) |
| philippine | 0.9202 (± 0.0076) | **0.9205 (± 0.0075)** | 0.9205 (± 0.0075) | 0.918 (± 0.0074) |
| phoneme | 0.9636 (± 0.0041) | 0.964 (± 0.0035) | **0.9641 (± 0.0033)** | 0.962 (± 0.0043) |
| porto-seguro | 0.6429 (± 0.0017) | **0.6432 (± 0.002)** | 0.6432 (± 0.002) | 0.6371 (± 0.0037) |
| qsar-biodeg | 0.9554 (± 0.0057) | **0.9559 (± 0.0064)** | 0.9559 (± 0.0062) | 0.9449 (± 0.0081) |
| riccardo | 0.9999 (± 0.0) | **0.9999 (± 0.0)** | 0.9999 (± 0.0001) | 0.9999 (± 0.0) |
| sf-police-incidents | 0.708 (± 0.0036) | **0.7085 (± 0.0033)** | 0.7085 (± 0.0034) | 0.6923 (± 0.007) |
| sylvine | **0.9928 (± 0.0013)** | 0.9928 (± 0.0013) | 0.9927 (± 0.0014) | 0.9922 (± 0.0016) |
| wilt | 0.9982 (± 0.0013) | **0.9982 (± 0.0012)** | 0.9982 (± 0.0013) | 0.9976 (± 0.0017) |

Table 11: **Validation ROC AUC - Multi-class**: The mean and standard deviation of the validation score over all folds for each method. The best methods per dataset are shown in bold. All methods close to the best method are considered best (using NumPy's default *isclose* function).

| Dataset | GES | QDO-ES | QO-ES | SingleBest |
|---|---|---|---|---|
| Diabetes130US | 0.7146 (± 0.0038) | **0.7147 (± 0.0038)** | **0.7147 (± 0.0038)** | 0.7101 (± 0.0024) |
| Fashion-MNIST | **0.9946 (± 0.0004)** | **0.9946 (± 0.0004)** | **0.9946 (± 0.0004)** | 0.9942 (± 0.0004) |
| GesturePhaseSegmentationProcessed | 0.9152 (± 0.0064) | 0.9153 (± 0.0064) | **0.9154 (± 0.0064)** | 0.9056 (± 0.0039) |
| KDDCup99 | **0.9999 (± 0.0001)** | 0.9999 (± 0.0001) | **0.9999 (± 0.0001)** | 0.9995 (± 0.0006) |
| amazon-commerce-reviews | 0.9959 (± 0.001) | **0.996 (± 0.001)** | 0.9959 (± 0.001) | 0.9878 (± 0.0032) |
| car | **1.0 (± 0.0)** | **1.0 (± 0.0)** | **1.0 (± 0.0)** | **1.0 (± 0.0)** |
| cmc | **0.7804 (± 0.0191)** | **0.7804 (± 0.0193)** | 0.7802 (± 0.0195) | 0.7665 (± 0.0197) |
| cnae-9 | 0.9992 (± 0.0003) | **0.9992 (± 0.0003)** | 0.9992 (± 0.0003) | 0.9985 (± 0.0006) |
| connect-4 | 0.9423 (± 0.0017) | **0.9426 (± 0.0017)** | 0.9425 (± 0.0017) | 0.9393 (± 0.0018) |
| covertype | **0.9991 (± 0.0001)** | **0.9991 (± 0.0001)** | **0.9991 (± 0.0001)** | 0.9989 (± 0.0001) |
| dilbert | **1.0 (± 0.0)** | **1.0 (± 0.0)** | **1.0 (± 0.0)** | 0.9999 (± 0.0) |
| dionis | **0.9993 (± 0.0001)** | 0.9993 (± 0.0001) | 0.9993 (± 0.0001) | 0.9987 (± 0.0003) |
| dna | 0.9962 (± 0.0009) | **0.9964 (± 0.0009)** | 0.9963 (± 0.0009) | 0.9954 (± 0.001) |
| eucalyptus | 0.9382 (± 0.0056) | **0.9386 (± 0.0054)** | 0.9384 (± 0.0057) | 0.928 (± 0.0056) |
| fabert | 0.9448 (± 0.0018) | **0.945 (± 0.0017)** | 0.945 (± 0.0017) | 0.9355 (± 0.0024) |
| first-order-theorem-proving | 0.8444 (± 0.0027) | 0.8449 (± 0.0025) | **0.8449 (± 0.0024)** | 0.8369 (± 0.0035) |
| helena | 0.8933 (± 0.0023) | **0.8935 (± 0.0021)** | 0.8935 (± 0.0021) | 0.8792 (± 0.0025) |
| jannis | 0.8838 (± 0.0023) | **0.884 (± 0.0024)** | 0.8839 (± 0.0025) | 0.8781 (± 0.003) |
| jungle_chess_2pcs_raw_endgame_complete | **0.993 (± 0.0015)** | 0.993 (± 0.0015) | 0.993 (± 0.0015) | 0.9918 (± 0.002) |
| mfeat-factors | **0.9998 (± 0.0002)** | 0.9998 (± 0.0002) | 0.9998 (± 0.0002) | 0.9996 (± 0.0004) |
| micro-mass | 0.9995 (± 0.0005) | 0.9995 (± 0.0005) | **0.9995 (± 0.0004)** | 0.9991 (± 0.0007) |
| okcupid-stem | 0.8287 (± 0.0021) | 0.8289 (± 0.0022) | **0.8289 (± 0.0022)** | 0.8259 (± 0.0018) |
| robert | 0.9009 (± 0.0035) | 0.9011 (± 0.0034) | **0.9011 (± 0.0035)** | 0.8903 (± 0.0032) |
| segment | 0.9975 (± 0.0007) | 0.9975 (± 0.0008) | **0.9975 (± 0.0008)** | 0.9967 (± 0.001) |
| shuttle | **1.0 (± 0.0)** | **1.0 (± 0.0)** | **1.0 (± 0.0)** | **1.0 (± 0.0)** |
| steel-plates-fault | 0.9712 (± 0.0047) | **0.9716 (± 0.0044)** | 0.9715 (± 0.0045) | 0.9688 (± 0.0041) |
| vehicle | **0.9779 (± 0.0044)** | 0.9779 (± 0.0044) | 0.9778 (± 0.0043) | 0.9734 (± 0.0047) |
| volkert | 0.9572 (± 0.0019) | **0.9573 (± 0.0017)** | 0.9572 (± 0.0019) | 0.9536 (± 0.0018) |
| wine-quality-white | 0.9064 (± 0.0075) | **0.9068 (± 0.0078)** | 0.9064 (± 0.0077) | 0.8982 (± 0.0072) |
| yeast | 0.8947 (± 0.0125) | **0.8961 (± 0.0137)** | 0.8956 (± 0.0134) | 0.8795 (± 0.0142) |

Table 12: **Validation Balanced Accuracy - Binary**: The mean and standard deviation of the validation score over all folds for each method. The best methods per dataset are shown in bold. All methods close to the best method are considered best (using NumPy's default *isclose* function).

| Dataset | GES | QDO-ES | QO-ES | SingleBest |
|---|---|---|---|---|
| APSFailure | 0.9718 (± 0.0033) | 0.9723 (± 0.0035) | **0.9727 (± 0.0036)** | 0.9663 (± 0.0048) |
| Amazon_employee_access | 0.8 (± 0.0071) | **0.8041 (± 0.0077)** | 0.804 (± 0.0074) | 0.7863 (± 0.0092) |
| Australian | 0.9113 (± 0.0106) | **0.9192 (± 0.0105)** | 0.9178 (± 0.0092) | 0.9049 (± 0.0127) |
| Bioresponse | 0.8079 (± 0.0126) | 0.8129 (± 0.0107) | **0.8137 (± 0.01)** | 0.7984 (± 0.0111) |
| Click_prediction_small | 0.6485 (± 0.0039) | 0.6489 (± 0.0048) | **0.6492 (± 0.0043)** | 0.6444 (± 0.0047) |
| Higgs | 0.7518 (± 0.0015) | **0.752 (± 0.0017)** | 0.752 (± 0.0017) | 0.7464 (± 0.0028) |
| Internet-Advertisements | 0.9664 (± 0.0052) | **0.9689 (± 0.005)** | 0.9688 (± 0.0068) | 0.9618 (± 0.0058) |
| KDDCup09-Upselling | 0.8112 (± 0.0057) | **0.8128 (± 0.0062)** | 0.8123 (± 0.0059) | 0.8052 (± 0.0054) |
| KDDCup09_appetency | **0.7749 (± 0.0157)** | 0.7744 (± 0.0152) | 0.7744 (± 0.0162) | 0.7636 (± 0.013) |
| MiniBooNE | 0.9475 (± 0.0014) | 0.9477 (± 0.0013) | **0.9478 (± 0.0013)** | 0.9449 (± 0.0016) |
| PhishingWebsites | 0.9754 (± 0.0019) | 0.9756 (± 0.0019) | **0.9764 (± 0.0018)** | 0.9733 (± 0.0018) |
| Satellite | 0.9916 (± 0.0114) | **0.9924 (± 0.0123)** | 0.9922 (± 0.0117) | 0.9844 (± 0.0182) |
| ada | 0.8535 (± 0.0083) | **0.8554 (± 0.0086)** | 0.8549 (± 0.0083) | 0.846 (± 0.0091) |
| adult | **0.8474 (± 0.0031)** | 0.8472 (± 0.003) | **0.8474 (± 0.0029)** | 0.8453 (± 0.0029) |
| airlines | 0.6634 (± 0.001) | 0.6636 (± 0.001) | **0.6638 (± 0.001)** | 0.658 (± 0.0012) |
| albert | 0.6931 (± 0.0018) | 0.6932 (± 0.0022) | **0.6933 (± 0.0021)** | 0.6897 (± 0.0017) |
| arcene | 0.9602 (± 0.0326) | **0.9699 (± 0.0227)** | 0.969 (± 0.0241) | 0.9437 (± 0.0339) |
| bank-marketing | 0.8781 (± 0.0044) | 0.8779 (± 0.0046) | **0.8782 (± 0.0044)** | 0.8765 (± 0.0042) |
| blood-transfusion-service-center | 0.73 (± 0.0227) | **0.7354 (± 0.0206)** | 0.7282 (± 0.0192) | 0.7078 (± 0.0168) |
| christine | 0.7794 (± 0.0063) | **0.784 (± 0.0071)** | 0.7825 (± 0.0071) | 0.7666 (± 0.0059) |
| churn | 0.9193 (± 0.0108) | 0.9204 (± 0.0102) | **0.9206 (± 0.01)** | 0.9144 (± 0.0103) |
| credit-g | 0.7727 (± 0.0204) | **0.7799 (± 0.0213)** | 0.7779 (± 0.0209) | 0.7571 (± 0.0219) |
| gina | **0.9775 (± 0.0038)** | 0.9773 (± 0.0037) | 0.9775 (± 0.0033) | 0.9733 (± 0.0055) |
| guillermo | 0.8421 (± 0.003) | **0.8441 (± 0.002)** | 0.844 (± 0.0028) | 0.8364 (± 0.0018) |
| jasmine | 0.8415 (± 0.0103) | **0.842 (± 0.0105)** | 0.8412 (± 0.0103) | 0.8273 (± 0.0079) |
| kc1 | 0.7852 (± 0.0153) | **0.7926 (± 0.0137)** | 0.7893 (± 0.0096) | 0.7696 (± 0.0152) |
| kick | 0.7094 (± 0.0034) | 0.7092 (± 0.004) | **0.71 (± 0.0039)** | 0.7033 (± 0.0038) |
| kr-vs-kp | 0.998 (± 0.0018) | **0.9982 (± 0.0013)** | 0.9982 (± 0.0015) | 0.9975 (± 0.0023) |
| madeline | 0.928 (± 0.0066) | **0.9294 (± 0.0057)** | 0.929 (± 0.0063) | 0.9207 (± 0.0086) |
| nomao | 0.9727 (± 0.0016) | 0.9726 (± 0.0017) | **0.9728 (± 0.0016)** | 0.9711 (± 0.0013) |
| numerai28.6 | 0.5268 (± 0.003) | 0.527 (± 0.0023) | **0.5272 (± 0.0028)** | 0.5234 (± 0.0025) |
| ozone-level-8hr | 0.9079 (± 0.0154) | **0.9139 (± 0.0089)** | 0.9102 (± 0.0155) | 0.9004 (± 0.0136) |
| pc4 | 0.9203 (± 0.015) | **0.9285 (± 0.01)** | 0.9281 (± 0.0117) | 0.9071 (± 0.0119) |
| philippine | 0.8479 (± 0.0103) | 0.8488 (± 0.0098) | **0.8494 (± 0.0093)** | 0.8395 (± 0.0103) |
| phoneme | 0.9019 (± 0.004) | **0.9042 (± 0.0039)** | 0.9035 (± 0.0036) | 0.8927 (± 0.0039) |
| porto-seguro | 0.6041 (± 0.0021) | 0.6037 (± 0.0019) | **0.6043 (± 0.0022)** | 0.6006 (± 0.0022) |
| qsar-biodeg | 0.9108 (± 0.011) | **0.9157 (± 0.009)** | 0.9143 (± 0.0101) | 0.8933 (± 0.0133) |
| riccardo | 0.999 (± 0.0004) | **0.999 (± 0.0004)** | 0.999 (± 0.0004) | 0.9987 (± 0.0005) |
| sf-police-incidents | 0.6376 (± 0.0038) | **0.6387 (± 0.0037)** | 0.6377 (± 0.0039) | 0.6252 (± 0.0055) |
| sylvine | 0.965 (± 0.0037) | 0.9654 (± 0.0028) | **0.9657 (± 0.0034)** | 0.963 (± 0.0038) |
| wilt | 0.9891 (± 0.0054) | **0.9907 (± 0.0055)** | 0.9894 (± 0.0056) | 0.9866 (± 0.0052) |

Table 13: **Validation Balanced Accuracy - Multi-class:** The mean and standard deviation of the validation score over all folds for each method. The best methods per dataset are shown in bold. All methods close to the best method are considered best (using NumPy's default *isclose* function).

| Dataset | GES | QDO-ES | QO-ES | SingleBest |
|---|---|---|---|---|
| Diabetes130US | **0.5045 (± 0.0019)** | 0.5041 (± 0.0021) | 0.5042 (± 0.0017) | 0.5002 (± 0.0022) |
| Fashion-MNIST | 0.9141 (± 0.0029) | **0.9154 (± 0.003)** | 0.915 (± 0.0027) | 0.9049 (± 0.0048) |
| GesturePhaseSegmentationProcessed | 0.6966 (± 0.0186) | **0.6971 (± 0.0187)** | 0.6971 (± 0.0197) | 0.6565 (± 0.0107) |
| KDDCup99 | 0.7938 (± 0.02) | **0.8072 (± 0.0223)** | 0.8026 (± 0.0208) | 0.7841 (± 0.0195) |
| amazon-commerce-reviews | 0.8868 (± 0.0166) | **0.8942 (± 0.0152)** | 0.8921 (± 0.0176) | 0.8337 (± 0.0159) |
| car | 0.9983 (± 0.0039) | **0.9994 (± 0.0017)** | 0.9992 (± 0.0017) | 0.9969 (± 0.0066) |
| cmc | 0.6256 (± 0.0167) | **0.6326 (± 0.0162)** | 0.6291 (± 0.015) | 0.6087 (± 0.0165) |
| cnae-9 | 0.9773 (± 0.0083) | **0.9782 (± 0.0079)** | **0.9782 (± 0.0076)** | 0.9711 (± 0.0085) |
| connect-4 | 0.7727 (± 0.0068) | **0.7733 (± 0.0066)** | 0.7731 (± 0.0064) | 0.7683 (± 0.0062) |
| covertype | 0.9641 (± 0.0016) | **0.9647 (± 0.0017)** | 0.9644 (± 0.0014) | 0.9553 (± 0.0011) |
| dilbert | 0.9973 (± 0.0008) | **0.9975 (± 0.0007)** | 0.9974 (± 0.0008) | 0.9953 (± 0.0019) |
| dionis | 0.8463 (± 0.0172) | **0.8487 (± 0.0159)** | 0.8476 (± 0.0164) | 0.8406 (± 0.0108) |
| dna | 0.9735 (± 0.0051) | **0.9763 (± 0.0035)** | 0.9749 (± 0.0047) | 0.9708 (± 0.0052) |
| eucalyptus | 0.7454 (± 0.0168) | 0.7467 (± 0.0083) | **0.7484 (± 0.0159)** | 0.7172 (± 0.0098) |
| fabert | 0.7306 (± 0.0047) | **0.7353 (± 0.0054)** | **0.7353 (± 0.0062)** | 0.71 (± 0.01) |
| first-order-theorem-proving | 0.5311 (± 0.008) | **0.5342 (± 0.0078)** | 0.5339 (± 0.0081) | 0.5178 (± 0.0071) |
| helena | 0.2777 (± 0.006) | **0.2819 (± 0.0066)** | 0.2812 (± 0.0073) | 0.2659 (± 0.0047) |
| jannis | 0.654 (± 0.0068) | **0.6549 (± 0.0063)** | 0.6545 (± 0.0066) | 0.648 (± 0.0058) |
| jungle_chess_2pcs_raw_endgame_complete | 0.9307 (± 0.0121) | **0.9314 (± 0.0124)** | 0.9311 (± 0.0123) | 0.923 (± 0.0115) |
| mfeat-factors | 0.9934 (± 0.0033) | 0.9934 (± 0.0035) | **0.9936 (± 0.0035)** | 0.9914 (± 0.0038) |
| micro-mass | **0.967 (± 0.0147)** | 0.9665 (± 0.0159) | 0.9659 (± 0.0162) | 0.9604 (± 0.0179) |
| okcupid-stem | 0.7027 (± 0.003) | 0.7032 (± 0.0024) | **0.7033 (± 0.0026)** | 0.6994 (± 0.0023) |
| robert | 0.5527 (± 0.0091) | **0.5564 (± 0.0099)** | 0.5554 (± 0.0104) | 0.5347 (± 0.0112) |
| segment | 0.9596 (± 0.0071) | **0.963 (± 0.0068)** | 0.9619 (± 0.0075) | 0.9511 (± 0.0075) |
| shuttle | **1.0 (± 0.0)** | **1.0 (± 0.0)** | **1.0 (± 0.0)** | **1.0 (± 0.0)** |
| steel-plates-fault | 0.8595 (± 0.0135) | **0.8633 (± 0.0118)** | 0.8633 (± 0.0126) | 0.8479 (± 0.0112) |
| vehicle | 0.902 (± 0.0111) | **0.9036 (± 0.0132)** | 0.9036 (± 0.012) | 0.8818 (± 0.014) |
| volkert | 0.7143 (± 0.0096) | **0.7162 (± 0.0085)** | 0.7157 (± 0.0074) | 0.6745 (± 0.0038) |
| wine-quality-white | 0.5782 (± 0.0453) | **0.599 (± 0.0274)** | 0.5901 (± 0.0296) | 0.5212 (± 0.0438) |
| yeast | 0.6134 (± 0.0265) | **0.6165 (± 0.0243)** | 0.6148 (± 0.0255) | 0.6017 (± 0.0309) |

