# OpenReview forum: "Q(D)O-ES: Population-based Quality (Diversity) Optimisation for Post Hoc Ensemble Selection in AutoML"
_automl.cc/AutoML/2023/Conference — AutoML 2023 MainTrack_

### Official Review · Reviewer_7tmR · 2023-04-13

**Potential Impact On The Field Of Automl Rating:** 3
**Technical Quality And Correctness Rating:** 3
**Clarity Rating:** 3

**Summary Of Contributions:**

This paper proposes two population based methods for post hoc ensemble selection in AutoML: QO-ES (quality optimization) and QDO-ES (quality diversity optimization). The general idea is to investigate whether the behavioural diversity of a population of ensembles may be beneficial for the performance. The methods are compared to greedy ensemble selection (GES) as well as the single best model. The authors claim that QO-ES and QDO-ES often outperform GES, as well as that behavioural diversity can be beneficial for post hoc ensembling.


**Actions Required To Increase Overall Recommendation:**

All in all, the negative aspects described under correctness and clarity need to be addressed to convince me to raise my score.


**Clarity:**

Overall, the paper is well structured and easily understood in most parts.

The mathematical formulation is mostly well done and consistent. $m$ could have been described within the text, to make the reader aware of its existence, and to avoid confusion later on when it is used. $r$ itself should be defined (not only $r(i)$). Some variables should be styled correctly, e.g., $r_{history}$ seems to be missing a \mathit{..}: $r_{\mathit{history}}$, nevertheless I am wondering why $h$ is not used for the history. Moreover, the diversity measures (ALC & CSS) should additionally be introduced with some formula.

The appendix is quite extensively used and referred to. Nevertheless, the main paper should cover all the necessary information to be understood. For E.g. tournament selection (line 179) should be briefly described before referring to the appendix. Moreover overfitting is discussed quite extensively, but the results thereof can only be found in the appendix.

Within the evaluation (around lines 235 and 250) it is not clear to what extent AutoSklearn und AutoGluon have been used.
Overall, captions should contain a short description of the experiment it belongs to.

There is probably a small grammar error in line 206, I would suggest: “We implemented three approaches initializing the set: [...]”

Lastly, the overall used CPU hours to perform the final evaluation of this paper would be of interest.


**Overall Review:**

## Positive

- Structure of the paper
- Most descriptions of methods and evaluation protocols
- Interesting idea of behavioural diversity

## Negative (see comments in other fields)

- Results and conclusions thereof are oversold
- Mathematical presentation could be improved


**Potential Impact On The Field Of Automl:**

The paper proposes two interesting methods for post hoc ensembling. Overall the experiments show that ensembling is usually better than the single best model, confirming already existing knowledge. Furthermore, Q(D)O-ES performs comparable to GES, making it a good alternative to GES. Additionally, experiments have shown that behavioural diversity can be beneficial, but also increases the risk of overfitting. Overall, the paper could have a medium impact to the field of AutoML.


**Review Confidence:**

4: You are confident in your assessment, but not absolutely certain. It is unlikely, but not impossible, that you did not understand some parts of the submission or that you are unfamiliar with some pieces of related work.

**Review Rating:**

8: Accept: Technically sound paper with major impact and strong evaluation, with perhaps some minor flaws.

**Review Summary:**

The paper proposes an interesting alternative to GES, although it is oversold in my opinion. It is well structured and described in most parts of the paper. Nevertheless, especially the results and discussions thereof could be improved and clarified. Additionally, I was wondering why only a single AutoML tool has been used for evaluation. Furthermore, in the beginning of the paper, I assumed that diversity would play a major role in this paper and would be further analyzed, but it turned out diversity was not so present.

**Technical Quality And Correctness:**

The authors have bold statements about the performance of their methods compared to GES, but none of the three ensembling methods investigated is significantly better than another on test data. Only on validation data a significant performance improvement of Q(D)O-ES can be detected over GES, which is a sign of overfitting. Therefore, the performance of Q(D)O-ES is oversold. E.g.: Figure 1 shows critical difference plots, where all ensembling selection methods are significantly better than the single best model, but none of them is significantly better than another. Still, the authors claim that Q(D)O-ES often outperforms GES, which is not true.

The three initialization approaches (lines 204 - 207) should all be defined mathematically. Furthermore, the definition of the first ensemble consisting only of a base model (line 206) is wrong, as it would result in an empty set.

Within Algorithm 2, line 7, the remark does not make sense to me as it is the same as on the left hand side.

The efficiency of Q(D)O-ES is discussed in lines 316 - 325, but there is no source given for those numbers. Any kind of table or plot would have been expected, distinguishing between the typical four cases in this paper.

Furthermore, an ablation study about which parts and parameters most affect the performance would be interesting. The mentioned method for assessing hyperparameter importance (Appendix F2) should be elaborated. Moreover, it is not distinguished, e.g., which initialization method has been used, and how important it is.

In general, I was wondering why the experiments have not been repeated for different seeds.

---

> ### Author Response · Authors · 2023-04-25
> **Response to Reviewer 7tmR (Part 2/2)**
>
> > “Lastly, the overall used CPU hours to perform the final evaluation of this paper would be of interest.“
>
> We did not state any time-related number w.r.t. our experiments in the manuscript, while it would indeed be beneficial. We will fix this in the updated manuscript. We estimated the wall clock time to be approximately 3.9 years (summed over all datasets, folds, metrics, and configurations; ignoring the parallelization of the ensemble method; we will provide details on this number in the Appendix of the updated manuscript). We did not capture the exact CPU time and cannot calculate this exactly in retrospect due to parallelization.
>
>
> > “In general, I was wondering why the experiments have not been repeated for different seeds.”
>
> We have used multiple folds (10 per dataset) following the AutoML benchmark. As the AutoML benchmark did not use seeds, we also did not look at different random seeds per fold. Moreover, considering the number of datasets and randomness induced by resampling via cross-validation, we believe our experiments already cover a good amount of randomisation.
>
> Furthermore, we initially deemed it too expensive to include seeds due to the large number of configurations. In retrospect, considering the estimated wall clock time of approximately 3.9 years, we believe this to be the case indeed.
>
> > “Within the evaluation (around lines 235 and 250) it is not clear to what extent AutoSklearn und AutoGluon have been used. Overall, captions should contain a short description of the experiment it belongs to.”
>
> We used Auto-Sklearn for the base models and only a pruning method from AutoGluon. This is a hyperparameter, as stated in Line 251. None of our experiments are specific to a pruning method. Like all other hyperparameters, the pruning method is selected by LOO CV and is not specific to certain experiments. Hence, we can not provide such descriptions in the captions. We will highlight this as part of the ablation study in the updated manuscript to further clarify this.
>
> &nbsp;
>
> We will try to resolve the remaining points you raised w.r.t. clarity in the updated manuscript. Thank you for the valuable and detailed feedback!

---

> ### Author Response · Authors · 2023-04-25
> **Response to Reviewer 7tmR (Part 1/2)**
>
>
> Dear Reviewer 7tmR,
>
> Thank you for your feedback. As we want to start a potential discussion early, given the short timeframe of the rebuttal, we are sharing our comments now and are stating below where we plan to adjust the manuscript. The updated manuscript will follow by the end of the week.
>
> In your review, you stated:
> > “The authors have bold statements about the performance of their methods compared to GES, but none of the three ensembling methods investigated is significantly better than another on test data. Only on validation data, a significant performance improvement of Q(D)O-ES can be detected over GES, which is a sign of overfitting. “
>
> We do agree that some of our statements, especially in the abstract, may be considered exaggerated, and we will fix this in the updated manuscript. Thank you very much for bringing this to our attention. However, in the result section, we believe that we provide a fair summary and comparison of the performance of the different methods. We also want to note that comparing only four algorithms via Nemenyi Post hoc tests results in requiring very strong performance differences (and therefore very stable mean ranks) to result in significant differences. Looking at the box plots, we can observe that Q(D)O-ES usually results in better relative performance in 3 out of the 4 scenarios (especially QO-ES showing strong relative performance)
>
> We also agree that the significant performance improvement on validation data is a sign of overfitting, as discussed in the result section. However, it is also a sign that Q(D)O-ES can better solve the problem based on the given data (i.e., on validation data) than GES, despite the solution’s performance not generalising to the test data. As the poor generalisation likely follows from Auto-sklearn’s validation method (33% hold-out split), we do believe the significant performance differences on validation data are important.
>
> > “Therefore, the performance of Q(D)O-ES is oversold. [...] Still, the authors claim that Q(D)O-ES often outperforms GES, which is not true.”
>
> The statement about ‘outperforms’ was related to mean rank and in relation to better relative performance, not in relation to significance. Since the mean rank of our methods is often better, it ‘outperforms’ GES, albeit not significantly. Especially considering the difference in relative performance (Figure 2). Our statement holds true but seems to have been misleading; we will add the point about non-significance in the abstract and clarify our statement in the introduction to avoid such confusion.
>
> > “The three initialization approaches (lines 204 - 207) should all be defined mathematically.“
>
> We will add formalized versions of the initialization method to the Appendix, likely in terms of pseudo-code, mirroring the current implementation.
>
> > “Furthermore, the definition of the first ensemble consisting only of a base model (line 206) is wrong, as it would result in an empty set.”
>
> Please note that the formula states that it is the set of multisets such that there exists one and only one element of $r$ that is $1$. As such, it can not result in an empty set but will always correspond to a set of one base model. Here, $\exists!$ means "exists exactly one" (e.g., see https://en.wikipedia.org/wiki/Uniqueness_quantification or Chapter 1 in [1]).
>
> > “Within Algorithm 2, line 7, the remark does not make sense to me as it is the same as on the left hand side. “
>
> The operation “<-” means assignment; while the mathematical operator “=” means equal. In other words, the right-hand side states that the output of `mutate()` is equal to the input $r_{sol}$. While the left-hand side only states that the output of `mutate()` is assigned as the new value of $r_{sol}$.
>
> > “Furthermore, an ablation study about which parts and parameters most affect the performance would be interesting. The mentioned method for assessing hyperparameter importance (Appendix F2) should be elaborated. Moreover, it is not distinguished, e.g., which initialization method has been used, and how important it is.”
>
> We will elaborate on the hyperparameter importance in the updated manuscript and provide an additional ablation study w.r.t. the values of hyperparameters. Thank you for pointing out the limitation of our current approach.
>
> &nbsp;
>
> References:
>
> [1] Howson, A. G. (1972). A handbook of terms used in algebra and analysis. Cambridge University Press.

---

> > ### Comment · Reviewer_7tmR · 2023-04-29
> > **Reviewer Response to Author Response**
> >
> > Dear Authors,
> >
> > thanks for the detailed response to my review and the adapted paper with highlighted changes. Most of my comments have been sufficiently addressed, except the following:
> >
> > **Figure 1: Critical Difference Plots**
> >
> > Within the caption the used dataset is missing (as it has been added to Figure 2)
> >
> > **Appendix D.7 Initialisation Approaches**
> >
> > * I am still quite sure that the formal definition of $S_1$ is not correct, let me elaborate: You only check $r(i)=1$, but you do not check that the others are $r(i')=0$. My suggestions are the following, alternatively you could check the sum of $r$:
> >     * $$ S_1 = \\{ (P,r) | ∃!𝑖 ∈ [m] : 𝑟(𝑖) = 1 \wedge \forall j \in [m], j \neq i: r(i)=0 \\} $$
> >     * $S_1 = \\{ (P,r) | \sum\limits_{i =1}^{m} r(i) =1\\}$.
> > * Similarly, I suggest the following for $S_2$:
> >     * $S_2 = \\{(𝑃, 𝑟) | 𝑟(𝑗) = 1 ∧ ∃!𝑖 ∈[m],  i \neq j:  𝑟(𝑖) = 1 \\}$
> >     * $S_2 = \\{(𝑃, 𝑟) | 𝑟(𝑗) = 1 ∧ \sum\limits_{i =1}^{m} r(i) =2 \\}$
> > * Furthermore, $S_3$ can also be defined by set notation: First define the set of all possible ensembles with 2 elements:
> >     * $\hat{S_3} = \\{ (P, r) | ∃!𝑖,j ∈ [m], i\neq j : 𝑟(𝑖) = 1 = r(j) \\}$
> >     * $S_3 \subset \hat{S_3}$ with $|S_3| = m$
> >
> > I will update my review rating (Weak Accept -> Accept), as I assume that these minor corrections will be done either before rebuttal ends, or latest for the camera ready version if the paper is accepted.

---

> > > ### Author Response · Authors · 2023-04-29
> > > **Thanks**
> > >
> > > Thanks a lot for you swift and constrcutive reply. We will definitly take a look at D.7 to make sure it is correct.
> > > Thanks for putting in the the time to even check the appendix in such a detailed manner!

---

> > > > ### Comment · Reviewer_7tmR · 2023-04-29
> > > > **Forgot one Point**
> > > >
> > > > I actually forgot one additional minor point:
> > > > * $\mathbb{Z}^+$ does not include $0$, therefore you will have a problem with $r(i)=0$, therefore you probably want to use $\mathbb{Z}^+_0$

---

> ### Author Response · Authors · 2023-04-28
> **Changes Addressing Your Concerns: Reviewer 7tmR**
>
> Dear Reviewer 7tmR,
>
> We have added several improvements to the manuscript to address your concerns. For a detailed list of all improvements, see our comment “The Updated Manuscript is Available” above.
>
> Based on your valuable and comprehensive feedback:
>
> * We adapted our somewhat exaggerated statements in the abstract and introduction that were overselling our contributions.
> * We improved the mathematical presentation for several components of our approach. Specifically, we formalised the diversity measures in Appendix D.1, and the initialisation approaches in Appendix D.7. Moreover, we were able to improve the overall mathematical notation for several aspects thanks to your feedback.
> * Likewise, we make sure to shortly explain something before pointing to the appendix for more details and moved the results for validation data from the appendix to the main paper.
> * Finally, we provided an estimate of the wall-clock time. Our calculations of this lower bound of the CPU time are explained in Appendix F.1.
>
> We believe these changes address your concerns and hope you consider increasing your rating based on the improved manuscript. Thank you again for your feedback.

---

### Official Review · Reviewer_wmVj · 2023-04-13

**Potential Impact On The Field Of Automl Rating:** 3
**Technical Quality And Correctness Rating:** 3
**Clarity:** This paper is well-orgnized.
**Clarity Rating:** 3
**Actions Required To Increase Overall Recommendation:** 1. Abalation studies, including the o…

**Summary Of Contributions:**

1. Using Quality-Diversity (QD) algorithms for Ensemble Selection (ES) problems.

2. Experimental results show the advantages of QDO-ES and QO-ES.

**Overall Review:**

Pros
1. The application of QD algorithms into ensemble selection is interesting and promising.

Cons

1. The experiment part is not solid enough. Please refer to "Actions Required To Increase Overall Recommendation".

**Potential Impact On The Field Of Automl:**

Automated ensemble selection is an important problem in AutoML.
Besides, the application of QD algorithms into ensemble selection is interesting. The researchers in QD will be interested in this work.

**Review Confidence:**

2: You are willing to defend your assessment, but it is quite likely that you did not understand the central parts of the submission or that you are unfamiliar with some pieces of related work.

**Review Rating:**

6: Borderline Leaning Accept: Technically sound paper where reasons to accept outweigh reasons to reject. Please use sparingly.

**Review Summary:**

This paper apply QD algorithms into ensemble selection problems. However, the experiment part should be further improved.

**Technical Quality And Correctness:**

This paper apply current QD algorithms into the ensemble selection problems to obtain a set of high-performing and diverse solutions.
However, the lacking of compared methods (except for greedy and single best), and the lacking of ablation studies weaken the quality.

---

> ### Author Response · Authors · 2023-04-25
> **Response to Reviewer wmVj**
>
> Dear Reviewer wmVj,
>
> Thank you for your feedback. As we want to start a potential discussion early, given the short timeframe of the rebuttal, we are sharing our comments now and are stating below where we plan to adjust the manuscript. The updated manuscript will follow by the end of the week.
>
> In your review, you state:
> > “This paper apply current QD algorithms into the ensemble selection problems to obtain a set of high-performing and diverse solutions..”
>
> We would like to provide additional clarification regarding this statement to avoid further confusion. While we are applying a QD-like algorithm with QDO-ES, QO-ES is not a QD algorithm. Our work in general focuses on population-based search, which is not only restricted to QD algorithms. In this sense, we constructed another competitor without making use of the concepts of QD, which is given by the QO-ES method.
>
> > “However, the lacking of compared methods (except for greedy and single best), and the lacking of ablation studies weaken the quality”
>
> We compared our approach to ensemble selection to the only other available and widely used method in AutoML, greedy ensemble selection (GES). We do not compare our approach to ensemble weighting methods (like stacking) since we are interested in efficient, i.e., sparse, ensembles, see background (Lines 98-100). Additionally, the contribution of this paper is an improvement of GES by extending it to a population-based search. Therefore, we believe comparing our methods only to GES is appropriate. Nevertheless, we agree that the current literature lacks a general comparison of post hoc ensembling methods; we believe this to be a promising avenue for future work.
>
> We would be grateful if you could provide us with references to other post hoc ensembling methods for AutoML that you would like us to compare. The referenced method that you provide in your review, DivBO [1], is not directly applicable to post hoc ensembling (which is concerned with constructing an ensemble itself given a set of models) but instead tackles the problem of how to efficiently find models in a CASH fashion to then construct an ensemble.
>
> In detail, DivBO [1] describes an approach to introduce diversity during CASH. However, our methods look at diversity during post hoc ensembling. The method discussed in [1] is comparable to the papers we have mentioned in the related work (Lines 116-120), which is very different from our approach.
>
> &nbsp;
>
> You mention in your review that the paper is well organised but rated it as “2: The work is not presented in a clear way in most parts.”. We would be willing to address any more concerns you have about the clarity.
>
> In “Actions Required”, you stated:
> > “More types of experiments are needed.”
>
> We compared our method on 71 datasets (10-fold cross-validation) across two metrics following the procedure of the AutoML benchmark, which is currently setting the standard for best practices in AutoML. To the best of our knowledge, no related work for ensemble selection or post hoc ensembling has ever done such an extensive comparison. Note that the estimated wall clock time of our current experiments is 3.9 years (we provide details on this in the Appendix of the updated manuscript). We think that adding more experiments would hinder the reproducibility of our results without adding substantial value to the paper.
>
> &nbsp;
>
> Regarding your comment, “Abalation studies, including the operators and the hyper-parameters.”. Thank you for the feedback, we note that we have the performance values for all tested configurations. We only used this for analyzing hyperparameter importance in the paper, but will include additional ablation studies in the updated manuscript.
>
> > “The verification (or visualization) of the diversity of obtained solutions should be discussed.”
>
> We agree that such an analysis would be interesting in principle. However, the goal of this work was not to foster diversity in the final solution but to achieve peak performance with the final solutions. In other words, our aim was to obtain high-performing solutions and not solutions that are diverse, as stated in the introduction. We admit our application of QDO is atypical; nevertheless, having access to a diverse population during optimisation can improve performance, even if one is only interested in a single best solution, as this was our main motivation.
> We provide a glimpse of the algorithmic “diversity” of base models in relation to the pruning methods TopN and SiloTopN in Appendix E.1, Table 1. Specifically, the columns regarding the mean number of distinct algorithms show that SiloTopN produces a much more algorithmically diverse set of base models than TopN. However, we did not reference this explicitly in Lines 306-308, which will be corrected  in the updated manuscript
>
> &nbsp;
>
> Thank you again for your time and valuable feedback!
>
> &nbsp;
>
> References
>
> [1] DivBO: Diversity-aware CASH for Ensemble Learning.

---

> ### Author Response · Authors · 2023-04-28
> **Changes Addressing Your Concerns: Reviewer wmVj**
>
> Dear Reviewer wmVj,
>
> We have added several improvements to the manuscript to address your concerns. For a detailed list of all improvements, see our comment “The Updated Manuscript is Available” above. Thank you for your feedback allowing us to improve our manuscript.
>
> We provided details on ablation studies w.r.t. the hyperparameters and their values (e.g., crossover operators) in Appendix F.3 and F.4; showing the importance of different components of our approach. Moreover, we now elaborate on the algorithmic diversity created by SiloTopN and point to values showing this algorithmic diversity per dataset in the appendix.
>
> We believe these changes address your concerns and hope you consider increasing your rating based on the improved manuscript.

---

> > ### Comment · Reviewer_wmVj · 2023-05-02
> > **Responses to authors**
> >
> > Thanks for your replies. I think the revised manuscript is better now and I will increase my clarity and overall score.

---

### Official Review · Reviewer_e8Qx · 2023-04-15

**Potential Impact On The Field Of Automl Rating:** 3
**Technical Quality And Correctness Rating:** 3
**Clarity Rating:** 2

**Summary Of Contributions:**

This paper studies post hoc ensemble election based on a population-based quality and diversity optimization for AutoML. The experimental results compared against greedy ensemble selection. The study found that diversity is one of the key factors to achieve a better performance for ensemble models.

**Actions Required To Increase Overall Recommendation:**

Answering the above questions and revising the paper accordingly would be beneficial for the AutoML community.

**Clarity:**

The paper is well-written, and it is easy to follow but the lack of detail in the experiment section makes it less clear.

**Overall Review:**

It is an interesting paper for the AutoML community, but further discussion and explanation of the evaluation and the results are needed to provide a comprehensive understanding.

**Potential Impact On The Field Of Automl:**

Model selection may critically change the performance of an AutoML platform, consequently advancing model selection may have a high impact on the field of AutoMLs. However, the lack of clarity on evaluation may decrease the impact of this study.

**Reproducibility (Optional):**

I appreciate that the authors provide the source-code of their work. Although I did not run the code, the code is clean and easy to follow.

**Review Confidence:**

3: You are fairly confident in your assessment. It is possible that you did not understand some parts of the submission or that you are unfamiliar with some pieces of related work.

**Review Rating:**

8: Accept: Technically sound paper with major impact and strong evaluation, with perhaps some minor flaws.

**Review Summary:**

The paper is well-written and easy to follow but lacks detail on experiment setup and evaluations that may impact comprehensiveness of the manuscript's findings.

**Technical Quality And Correctness:**

I enjoyed reading this interesting paper. Although the proposed approach and recommendations are not a novel idea (see Nam et al. [1]), the paper has promising empirical results and sharing these results with the AutoML community is beneficial. However, the paper has several issues as follows that need to be addressed and discussed.

In Line 227, the authors stated that “...we observe that for all scenarios, post hoc ensemble selection is statistically significantly better than the single best model ...” but it raises several important questions which are not answered in this study, such as: what is the cost of this improvement? How much additional time is required per dataset to achieve this performance gain? (There is only a brief description in Line 321). Adding ensemble models are expensive compared to a single model. How does the proposed approach handle large datasets? My understanding of the current experimental setup is that model selection applies to only 10% of data for each dataset in each iteration, which makes it feasible to have diverse ensemble algorithms.


In addition, the authors emphasize that their report is based on a 10-fold cross-validation, which is not clear whether the performance is based on average of selected ensemble model in each iteration or if it is based on the performance of selected ensemble on the whole dataset.

Furthermore, it would be beneficial to have fine-grained evaluation results per dataset for a more comprehensive analsyis.

Minor issues:

- Choose either UK English or US English and make it consistent throughout the whole manuscript (i.e., “Optimization” vs “optimisation” or “normalised” vs “normalized”)


[1] Nam, Giung, et al. "Diversity matters when learning from ensembles." Advances in neural information processing systems 34 (2021): 8367-8377.

---

> ### Author Response · Authors · 2023-04-25
> **Response to Reviewer e8Qx**
>
> Dear Reviewer e8Qx,
>
> Thank you for your feedback. As we want to start a potential discussion early, given the short timeframe of the rebuttal, we are sharing our comments now and are stating below where we plan to adjust the manuscript. The updated manuscript will follow by the end of the week.
>
> In your review, you state:
> > “Although the proposed approach and recommendations are not a novel idea (see Nam et al. [1]),”.
>
> It is true that the idea of using diversity in ensembling is not novel. However, to the best of our knowledge, we are the first to propose and evaluate population-based algorithms for greedy ensemble selection and AutoML, as discussed in the related work section.
>
> Regarding the paper of Nam et al. [1], we believe that it is not directly related to our work. In contrast, the papers on diversity mentioned in our introduction (Lines 45-57) are more closely related papers. Nam et al. [1] discuss a model distillation approach motivated by ensemble diversity literature. Model distillation aggregates model parameters or behavior to build one model. In contrast, ensembles aggregate the predictions of multiple models. In our work, we talk about a post hoc ensembling method motivated by ensemble diversity.
>
> > “In Line 227, the authors stated [...] but it raises several important questions which are not answered in this study, such as: what is the cost of this improvement? How much additional time is required per dataset to achieve this performance gain? (There is only a brief description in Line 321). Adding ensemble models are expensive compared to a single model How does the proposed approach handle large datasets?.”
>
> We assume you are referring to Line 277. We agree that this raises the questions you have mentioned, but most prominent AutoML systems use post hoc ensembling, and it is not the primary goal of this study to evaluate how much cost is induced by it. However, we show that our method does not incur significantly higher costs than the state-of-the-art method GES.
> Moreover, as our approaches function similarly efficiently to GES (Line 321) and handles large datasets similarly well to GES, answering questions related to the general efficiency of post hoc ensembling was not a goal of our paper.  We note that post hoc ensembling methods are in general efficient even for larger datasets, as AutoML systems store the validation predictions [2]. Multiple large datasets are part of the AutoML benchmark datasets, see Table 1 in the Appendix. Regarding this point and your later point “Furthermore, it would be beneficial to have fine-grained evaluation results per dataset for a more comprehensive analysis.” – our code provides the results per dataset, but these were not in the manuscript. We will provide an overview of performance per dataset in the updated manuscript.
>
> > “My understanding of the current experimental setup is that model selection applies to only 10% of data for each dataset in each iteration, which makes it feasible to have diverse ensemble algorithms.”
>
> In post hoc ensemble selection, CASH is not performed. Instead, post hoc ensemble selection is concerned with constructing an ensemble given a set of models. In our experiments, Auto-Sklearn’s model selection is performed on all data (using a hold-out split for validation). Our ensemble methods are then trained on the validation data from the hold-out split. By default and in our experiments, this validation data corresponds to 33% of the training data. We noticed that we only stated this in Line 295. For clarity, we will state this earlier in Section 5 in the updated manuscript.
>
> > “In addition, the authors emphasize that their report is based on a 10-fold cross-validation, which is not clear whether the performance is based on average of selected ensemble model in each iteration or if it is based on the performance of selected ensemble on the whole dataset.”
>
> Our evaluation protocol follows the AutoML benchmark. Consequently, the used datasets were split into 10 folds for cross-validation. For each split, we compute the score of each ensemble method and then average these fold scores. A method's fold score is the final ensemble's performance for the whole dataset. We did not evaluate the performance over time or iterations.
>
> &nbsp;
>
> Besides our comments above, we will update the explanations of the experimental setup and evaluations to increase clarity in the updated manuscript. Thank you again for your valuable feedback.
>
> References
>
> [1] Nam, Giung, et al. "Diversity matters when learning from ensembles." Advances in neural information processing systems 34 (2021): 8367-8377.
>
> [2] Feurer, M., Klein, A., Eggensperger, K., Springenberg, J., Blum, M., & Hutter, F. (2015). Efficient and Robust Automated Machine Learning. Advances in Neural Information Processing Systems, 28.

---

> > ### Comment · Reviewer_e8Qx · 2023-05-02
> > **Revised Version**
> >
> > I thank the authors for revising the manuscript to address my concerns. Although not all major issues have been resolved, in my opinion, the paper's presentation and experiments are more convincing. I have adjusted my rating accordingly.

---

> ### Author Response · Authors · 2023-04-28
> **Changes Addressing Your Concerns: Reviewer e8Qx**
>
> Dear Reviewer e8Qx,
>
> We have added several improvements to the manuscript to address your concerns. For a detailed list of all improvements, see our comment “The Updated Manuscript is Available” above.
>
> We now provide more details on the efficiency of all compared approaches in the result section. Specifically, stating the additional time added on average on top by running the methods for $50$ iterations. Moreover, we clarified what data is used during model selection in Section 5. Furthermore, we provided fine-grained evaluation results per dataset in Appendix H and made our usage of UK English consistent. Thank you for raising these concerns in your review.
>
> We believe these changes address your concerns and hope you consider increasing your rating based on the improved manuscript.

---

### Review · Reproducibility_Reviewer_gdJf · 2023-04-21

**Completeness Of Code And Dataset Supplement Rating:** 4
**Usability And Ease Of Reproducibility Rating:** 4
**Actions Required To Increase The Reproducibility And Overall Recommendation:** Everything is done well enough.

**Completeness Of Code And Dataset Supplement:**

The code and dataset are complete.

**Overall Reproducibility Review:**

Code runs with minimal effort, instructions are clean and the results clearly appear in experiments logs. May be it would be easier to use a single docker image with all the requirements for all experiments.

**Review Confidence:**

3: You are fairly confident in your assessment. It is possible that you did not understand some parts of the submission or that you are unfamiliar with some pieces of the code or data.

**Review Rating:**

9: Strong Accept, all aspects of this are easily reproducible.

**Review Summary:**

The supplemented code is working. Reproducibility does not raise any questions.

**Summary Of Necessary Code And Dataset Supplement:**

The authors had to implement different model ensembling methods and evaluate them on AutoML benchmark. The datasets are accessible via OpenML.

**Usability And Ease Of Reproducibility:**

The code is clean and instructions are complete and correct. Due to lack of computing resources, I did not reproduce the results in full, but the minimal example worked correctly. The presented results are not in doubt.

---

### Official Review · Reviewer_JBRh · 2023-04-24

**Potential Impact On The Field Of Automl Rating:** 2
**Technical Quality And Correctness:** The paper sounds good technically and…
**Technical Quality And Correctness Rating:** 3
**Clarity Rating:** 3

**Summary Of Contributions:**

This paper introduces two new population-based ensemble selection methods in post hoc optimization(QO-ES and QDO-ES) for AutoML, a version of ensemble selection where the search of optimism is not based on greedy search but based on population optimization. The authors present a background of ensemble learning and the motivations to pursue this work on using population-based methods instead of greedy methods. The new approach was explained in section 4 and provides guidance for each step of the proposed method.

**Actions Required To Increase Overall Recommendation:**

See overall review and adress comments.
The paper needs to be reorganized to show better the potential of the proposed approach.

**Clarity:**

The authors may need to introduce the method in a more organized way. especially section 4.

**Overall Review:**

The paper has several notable strengths:
     The authors tackle a practical use case and propose a straightforward solution that is effective and efficient. The proposed solution addresses the problem of post hoc ensemble selection methods in AutoML.
     The paper is well-written and in an easily understandable manner.
     The results are well-visualized and straightforward to interpret.

However, there are some negative aspects to the paper:
     The proposed approach is not well presented
     It is not very clear how the method could influence on the decision making part in AutoML
     The results representation /explanation is a bit complicated to follow and need to be improved.
     The approach need to be tested on more benchmarks to make intuitive conclusions.

**Potential Impact On The Field Of Automl:**

The proposed methods investigate a new version of ensemble selection optimization methods for AutoML, which assumes methods using mutation and crossover operators, QO-ES gradually enhances the top-performing ensembles from a repository it keeps. With the use of techniques from quality diversity optimization, QDO-ES expands upon QO-ES while also preserving an archive of behaviorally varied ensembles throughout the optimization process.
The experimental results demonstrate that the proposed approaches are methods to obtain good results. However, the potential impact of the solution is low but still offers meaningful insights.

**Review Confidence:**

3: You are fairly confident in your assessment. It is possible that you did not understand some parts of the submission or that you are unfamiliar with some pieces of related work.

**Review Rating:**

5: Borderline Leaning Reject: Technically sound paper where reasons to reject nonetheless outweigh reasons to accept. Please use sparingly.

**Review Summary:**

The suggested problem setting is intriguing and offers a fresh perspective on post hoc ensemble selection. However, the suggested strategy is rather straightforward, and the authors offer no firm recommendations. replacing a greedy search with a population-based method to predict higher performance but still not conclusive to make better decisions for AutoML.

---

> ### Author Response · Authors · 2023-04-25
> **Response to Reviewer JBRh**
>
> Dear Reviewer JBRh,
>
> Thank you for your feedback.
>
> Following your remarks, we are working on ways to improve the clarity of Section 4. However, to allow for discussions, we decided to answer the rest of your comments sooner rather than later
>
> You mentioned that our approach should be tested on more benchmarks.
> We compared our method on 71 datasets (10-fold cross-validation) across two metrics following the procedure of the AutoML benchmark, which is currently setting the standard for best practices in AutoML. To the best of our knowledge, no related work for ensemble selection or post hoc ensembling has ever done such an extensive comparison. Note that the estimated wall clock time of our current experiments is approximately 3.9 years (we provide details on this number in the Appendix of the updated manuscript). We think that adding more experiments would hinder the reproducibility of our results without adding substantial value to the paper.
>
> You state:
> > “It is not very clear how the method could influence on the decision making part in AutoML”.
>
> Sadly, it is slightly unclear to us what exactly you are referring to here. Our methods do not directly influence the decision making part of AutoML (i.e., CASH) but are a post-processing step, i.e., post-hoc ensemble selection. Could you please elaborate on this such that we can properly reply?
>
> Moreover, you state:
> > “However, the potential impact of the solution is low but still offers meaningful insights.”
>
> In the introduction, we listed many AutoML systems using post hoc ensembling, specifically greedy ensemble selection (GES). Because our method improves over GES, it can potentially impact most AutoML systems currently in use.

---

> ### Author Response · Authors · 2023-04-28
> **Changes Addressing Your Concerns: Reviewer JBRh**
>
> Dear Reviewer JBRh,
>
> Our updated manuscript contains several changes to improve clarity and notational quality.
> Additionally, the Appendix now includes several formalisations of components of our approach, which aim to resolve the unclear aspects of Section 4. Thank you for pointing this out to us.
>
> Moreover, we formulate an ablation study to provide conclusive data to make better decisions for AutoML.
>
> We believe these changes address your concerns w.r.t. clarity and conclusiveness and hope you consider increasing your rating based on the improved manuscript.
>
> For a detailed list of all improvements, see our comment “The Updated Manuscript is Available” above.

---

### Author Response · Authors · 2023-04-27
**Updated Manuscript Available**

Dear Reviewers,

We updated the manuscript taking into account your valuable feedback. *All changes to the manuscript are coloured in green*.

In detail, we changed the following major points:

* We estimated the wall-clock time of our experiments, stated this in the result section, and provided details on the estimation in Appendix F.1.
* In the result section, we expanded our discussion on ensemble efficiency and provided an overview in Table 1.
* We highlight that performance differences are only statistically significant on validation data, although Q(D)O-ES often outrank GES on both validation and test data. We further include validation results in Figure 3, showing that our approaches strongly outperform GES on validation data. The claim that our approaches perform well is further backed up by the performance overviews per dataset in Appendix H.
* We mention the algorithmic diversity created by SiloTopN explicitly in the result section and point to Table 2 in the Appendix showing the algorithmic diversity per dataset.
* We provide formalisations of the diversity measures in Appendix D.1 and formalisations of the initialisation approaches in Appendix D.7.
* We touch up ablation studies w.r.t. components of Q(D)O-ES and the configurations of these components in the result section. These ablation studies are detailed in Appendix F.3 and F.4.
* EDIT-1: We addressed the additional comments of reviewer 7tmR regarding the new formal definitions and Figure 1.

Additionally, we changed other minor things to improve clarity and notational quality. Moreover, we add all new code related to these changes to Anonymous GitHub.

We believe these changes address the concerns raised by you. Therefore, we politely ask you to verify if your concerns have been resolved and reconsider your rating.

Thank you for your time and valuable feedback. We are happy to participate in further discussions and willing to resolve additional concerns.